# Emergent Kitaev materials in synthetic Fermi-Hubbard bilayers

Daniel González-Cuadra[1, 2, *] and Alejandro Bermudez[3]

[1]*Department of Physics, Harvard University, Cambridge, MA 02138, USA*
[2]*Institute of Fundamental Physics IFF-CSIC, Calle Serrano 113b, 28006 Madrid, Spain*
[3]*Instituto de Física Teórica UAM-CSIC, Universidad Autónoma de Madrid, Cantoblanco, 28049, Madrid, Spain*

We investigate the emergence of bond-directional spin-spin interactions in a synthetic Fermi-Hubbard bilayer that can be realized with ultracold fermions in Raman optical lattices. The model exploits synthetic dimensions to couple two honeycomb layers, each corresponding to a different hyperfine atomic state, via Raman-assisted tunneling and, moreover, via an inter-layer Hubbard repulsion due to the cold-atom scattering. In the strong-coupling regime at half filling, we derive effective spin Hamiltonians for the kinetic exchange featuring Kitaev, Heisenberg, off-diagonal exchange (Γ-couplings), as well as tunable Dzyaloshinskii–Moriya interatcions. We identify specific configurations that generate both ferromagnetic and antiferromagnetic Kitaev couplings with various perturbations of relevance to Kitaev materials, providing a tunable platform that can explore how quantum spin liquids emerge from itinerant fermion systems. We analyze the Fermi-liquid and Mott-insulating phases, highlighting a correspondence between Dirac and Majorana quasi-particles, with possible phase transitions thereof. In an extreme anisotropic limit, we show that the model reduces to an inter-layer ribbon in a quasi-1D ladder, allowing for a numerical study of the correlated ground state using matrix product states. We find a transition from a symmetry-protected topological insulator to a Kitaev-like regime characterized by non-local string order. Our results establish that cold-atom quantum simulators based on Raman optical lattices can be a playground for extended Kitaev models, bridging itinerant fermionic systems and spin-liquid physics.

## CONTENTS

## I. INTRODUCTION

Kitaev's honeycomb model [1], which involves a set of spins arranged on a honeycomb lattice interacting via anisotropic bond-dependent spin-spin couplings [2, 3], has become a cornerstone for our current understanding of quantum spin liquids (QSLs) [4] and topological order (TO) [5]. In particular, it provides a clean playground to explore the phenomenon of fractionalization, as the low-energy quasiparticles of this spin-1/2 system cannot be described in terms of its elementary constituents, contrasting the case of magnons in more standard magnetic materials. In Kitaev's model, the quasiparticles fractionalize into charged-neutral spinons described by Majorana fermions [6], and vortex-like excitations of an emergent gauge field, which are commonly known as visons [7]. These excitations, whose combination forms the elementary spins, become deconfined in the QSL phase. In its original incarnation, Kitaev's model can be solved exactly by its mapping to a Majorana fermion model under a background $\mathbb{Z}_2$ gauge field [1, 8]. This has allowed for rigorous derivations showing how this state of matter connects explicitly to the expected characteristic features of a QSL. This includes the vanishing of spin-spin correlations beyond nearest neighbors [9], which showcase the absence of long-range magnetic ordering in favor of long-range entanglement patterns quantified by the topological entanglement entropy [10], and the appearance of either Abelian or non-Abelian anyonic quasiparticles depending on the microscopic couplings [1]. Deep in the gapped phases of Kitaev's model, the low-energy properties can be mapped onto those of Kitaev's toric code [11, 12], a seminal work that connects topological codes and quantum error correction [13] with $\mathbb{Z}_2$ lattice gauge theories and deconfinement [14].

Some years after Kitaev's seminal work, Jackeli and Khaliullin proposed that these QSLs can be realised in certain transition-metal compounds [15]. The candidate materials must display strong spin-orbit coupling, crystal-field effects specific of a certain lattice geometry, and Mott insulating behavior resulting from a strong electron-electron repulsion. Since then, various so-called Kitaev materials have been explored experimentally [16–18], coming to the conclusion that Kitaev's bond-dependent interactions typically complete with other microscopic terms in most realistic situations. This includes the typical Heisenberg interactions for kinetic exchange in Mott insulators [19], which tend to favor other magnetic phases [20–22]. In addition, one should also consider the non-zero temperatures of experiments, which can lie be-

arXiv:2504.15755v1 [cond-mat.quant-gas] 22 Apr 2025

* dgonzalezcuadra@fas.harvard.edu

yond the temperature below which one would find the Kitaev's QSLs, favoring instead a disorder paramagnet, or standard magnetic orders when the additional spin-spin couplings and magnetic fields are considered [3].

Contrary to the pristine Kitaev model, the full spin model of these Kitaev's material is not exactly solvable, and poses a complicated quantum many-body problem [16, 18]. In fact, the interplay of the bond-dependent Kitaev frustration, the additional spin-spin interactions and the non-zero temperature, leads to a sign problem for Monte Carlo simulations, as we now discuss. In the ideal case, the mapping to Majorana fermions can be used to develop a sign-free Markov-chain Monte Carlo sampling over the $\mathbb{Z}_2$ gauge field configurations, allowing to efficiently explore finite temperatures [23–27]. In contrast, when the gauge-field configurations are not conserved, e.g. arbitrary perturbations or problems involving real-time dynamics, a different strategy is required such as the continuous-time quantum Monte Carlo approach of [28–30]. This approach, however, is afflicted by a severe sign problem when lowering the temperature to the scales of interest. Some strategies to alleviate this sign problem have also been discussed more recently [31, 32]. A sign-error-free alternative is the use of variational tensor-network algorithms [33, 34], such as the density-matrix renormalization group for Kitaev inter-layer ribbons, cylinders or tori [35–39]. Alternatively, 2D variational approaches may be pursued such as infinite projected entangled pairs and tensor product states, addressing the thermodynamic limit [35, 40–47]. With the exception of [43], tensor-network approaches have not explored thermal effects, and also face challenges regarding efficient contraction [48] or entanglement growth in dynamics [49].

In this article, we explore how Kitaev materials can be investigated instead using quantum simulators [50, 51]. These are controllable quantum devices that can be 'programmed' such that they behave according to a specific model under investigation, allowing us to access regimes that cannot be simulated using traditional methods. In particular, here we focus on the emergence of Kitaev physics from an underlying multi-orbital Fermi-Hubbard model. Here, a Fermi liquid gives way to a Kitaev-Mott insulator as the Hubbard interactions are increased, which actually shows competing quantum spin liquid(s) and other more standard magnetic or disordered phases. We are not only interested in Kitaev's physics, but in understanding the emergence process quantitatively, a problem that remains largely unexplored to the best of our knowledge. This is arguably a more complicated many-body problem, as it deals with itinerant strongly-correlated fermions requiring additional energy scales. To progress in this direction, it would be highly desirable to find a minimal Fermi-Hubbard model that can encompass this rich Kitaev physics in the Mott-insulating limit, albeit dispensing with the various intricacies and limitations of the real transition-metal compounds mentioned above. In this context, analog quantum simulation proposals have, thus far, mainly focused on the implementation of the pristine Kitaev spin model. These include ultra-cold atoms in state-dependent optical lattices [52], cold polar molecules [53–55], trapped-ion crystals in micro-fabricated surface traps [56], superconducting

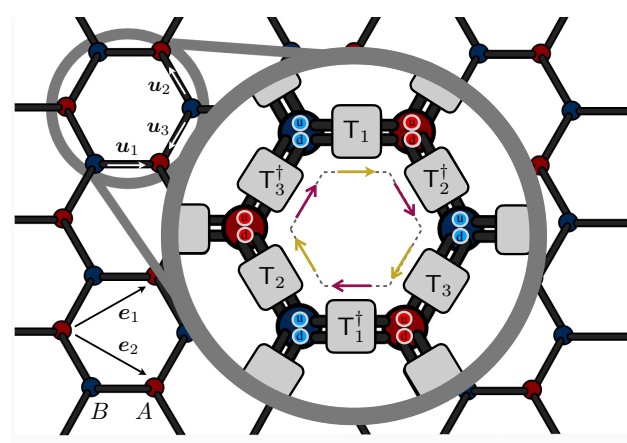

FIG. 1. **Fermi-Hubbard bilayer (FHB):** Sketch of the synthetic model defined on two vertically stacked honeycomb layers, each with a two-site unit cell $A$ and $B$, and two (three) unit (bond) vectors $\boldsymbol{e}_1, \boldsymbol{e}_2$ ($\boldsymbol{u}_1, \boldsymbol{u}_2, \boldsymbol{u}_3$). In addition to fermionic tunnelings $\mathsf{T}_i$, we consider on-site interaction $U$ coupling the two layers. Each lattice site hosts two fermionic states (u, d), and the atoms can tunnel between neighboring sites within each layer via complex, bond-dependent tunnelings.

circuits [57], and Rydberg tweezer arrays with laser-assisted dipole interactions [58]. In the spirit of digital quantum simulations [59], recent proposals have exploited Floquet engineering to achieve the bond-dependent model by a periodic sequence of isotropic spin-spin couplings interspersed with single-spin rotations [60, 61], leading to the first experimental realizations of the non-abelian spin liquid phase in the solvable Kitaev model [62, 63]. Finally, there have also been theoretical studies of variational quantum eigensolvers for Kitaev's model [64–66].

These works have not focused on the extended class of models for Kitaev materials, nor on the quantitative emergence of Kitaev's physics as one gradually enters in the strongly-interacting regime. The goal of this work is to fill in this gap by exploring a synthetic Fermi-Hubbard bilayer (FHB) with intra- and inter-layer tunnelings, and an inter-layer density-density repulsion. We show that, in the strongly-interacting half-filled limit, one obtains a Kitaev-Mott insulator in which the typical kinetic exchange couplings characteristic of Kitaev materials can be designed, and their relative strengths controlled. This minimal model avoids many of the intricacies of transition-metal Kitaev material candidates [16–18]. For such compounds, the appearance of Kitaev terms depends on the point-group symmetries of the solid-state matrix, the spin-orbit coupling, and the Hubbard's and Hund's interactions. Their interplay is responsible, for instance, for the appearance of effective pseudo-spin-1/2 doublets arising from a larger manifold of multiplets and, more importantly, for how the Kitaev couplings appear as a result of higher-order super-exchange processes involving the Hund's coupling to higher-pseudo-spin orbitals [16]. This mechanism can be affected by distortions of the crystal field, by a larger extent of the orbitals, or by contributions arising from other higher-lying orbitals. Instead, as we show below, our synthetic FHB

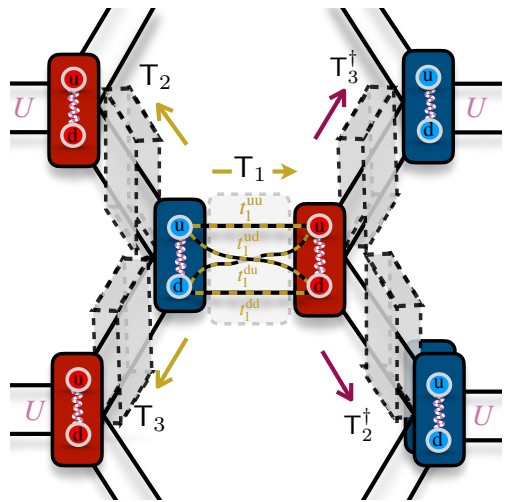

FIG. 2. **intra- and inter-layer tunnelings:** Illustration of the different types of tunneling processes used to construct bond-dependent exchange interactions. The tunneling matrices $\mathsf{T}_i$ and their time-reversed counterparts $\mathsf{T}_i^\dagger$ allow for encoding a bond-directional hopping, which has inter-layer $t_i^{\mathrm{uu}}, t_i^{\mathrm{dd}}$ and intra-layer $t_i^{\mathrm{ud}}, t_i^{\mathrm{du}}$ tunnelings.

provides a much simpler playground with a single orbital per site and layer that is amenable of being implemented in experiments of ultra-cold fermions in Raman lattices [67–70]. While prior quantum simulators focused on directly simulating Kitaev spin models, we aim at understanding the onset of bond-directional interactions from itinerant fermions as they transition into a Mott regime by tunning the microscopic parameters of the cold-atom quantum simulator.

This article is organized as follows. In Sec.II, we introduce the synthetic Fermi-Hubbard bilayer, describing its lattice geometry, tunneling structure, and interaction terms. We then perform a strong-coupling expansion, deriving the effective spin Hamiltonian and analyzing the conditions under which Kitaev, Heisenberg, and other anisotropic interactions emerge in a Kiatev-Mott insulating regime. We turn to a numerical analysis in Sec. III, where we explore the model on an inter-layer ribbon geometry using matrix product state (MPS) methods, revealing signatures of topological order and one-dimensional precursors to two-dimensional spin liquids driven by increasing Hubbard interactions. In Sec. IV, we describe a scheme for a possible experimental realization using fermionic atoms in Raman optical lattices, emphasizing the requirements for engineering the necessary tunneling matrices. We conclude and provide an outlook in Sec. V.

## II. SYNTHETIC FERMI-HUBBARD BILAYER

### A. The bilayer honeycomb model

Let us introduce the model under study. We consider a honeycomb bilayer with fermions residing on the corresponding two-site unit cells and being created and annihilated by $a_\ell^\dagger(\boldsymbol{r_n}), b_\ell^\dagger(\boldsymbol{r_n}), a_\ell(\boldsymbol{r_n}), b_\ell(\boldsymbol{r_n})$. Here, the $(A, B)$ unit cells

$\boldsymbol{r_n} = n_1 \boldsymbol{e}_1 + n_2 \boldsymbol{e}_2$ are labeled by the integers $\boldsymbol{n} = (n_1, n_2) \in \mathbb{Z}_{N_1} \times \mathbb{Z}_{N_2}$, where we make use of the unit vectors $\boldsymbol{e}_1 = \frac{a}{2}(1, \sqrt{3}), \boldsymbol{e}_2 = \frac{a}{2}(1, -\sqrt{3})$ with $a$ being the lattice spacing. The index $\ell = \mathrm{u}, \mathrm{d}$ labels the upper and lower layers, as depicted in Fig 1. In this honeycomb geometry, the $A(B)$ fermions can tunnel to the $B(A)$ neighbors via both intra- and inter-layer processes. Introducing Nambu spinors $\psi_A(\boldsymbol{r_n}) = (a_\mathrm{u}(\boldsymbol{r_n}), a_\mathrm{d}(\boldsymbol{r_n}))^\mathrm{t}, \psi_B(\boldsymbol{r_n}) = (b_\mathrm{u}(\boldsymbol{r_n}), b_\mathrm{d}(\boldsymbol{r_n}))^\mathrm{t}$ for each sublattice, the bilayer tunneling can be expressed as

$$H_0 = \sum_{i=1,2,3} \sum_{\boldsymbol{n}} \psi_A^\dagger(\boldsymbol{r_n} + \boldsymbol{u}_i)\, \mathsf{T}_i\, \psi_B(\boldsymbol{r_n}) + \mathrm{H.c.}, \quad (1)$$

where we set $\hbar = 1$ in the rest of the manuscript. In this expression, we have made use of the three bond vectors $\boldsymbol{u}_1 = a(1, 0), \boldsymbol{u}_2 = \frac{a}{2}(-1, \sqrt{3}), \boldsymbol{u}_3 = \frac{a}{2}(-1, -\sqrt{3})$, and defined the generic bond-dependent tunneling matrices

$$\mathsf{T}_i = \begin{pmatrix} t_i^{\mathrm{uu}} & t_i^{\mathrm{du}} \\ t_i^{\mathrm{ud}} & t_i^{\mathrm{dd}} \end{pmatrix} : \quad \mathsf{T}_i \in \mathrm{GL}(2, \mathbb{C}). \quad (2)$$

Therefore, $t_i^{\ell\ell'} \in \mathbb{C}$ is the tunneling strength along the $i$-th bond from the $\ell$-th layer to the $\ell'$-th one, as depicted in Fig. 2. This tight-biding model can be seen as a generalization of bilayer graphene with AA stacking [71]. For instance, setting $t_i^{\mathrm{du}} = t_i^{\mathrm{ud}} = 0$, and $t_i^{\mathrm{uu}} = t_i^{\mathrm{dd}} = -t \in \mathbb{R}$ would yield two copies of graphene's band structure, leading to a Dirac semi-metal for the non-interacting groundstate. Our more flexible tunneling matrix presents off-diagonal terms reminiscent of the so-called Rashba and Dresselhaus spin-orbit coupling [72, 73], provided the layer index is interpreted as the electronic spin. The analogy is however not exact, as we allow for a generic linear transformation over the field of complex numbers (2). The complex nature of the tunnelings may also suggest a connection to non-Abelian background gauge fields [74–76], in particular $\boldsymbol{A}(\boldsymbol{r}_n) \in \mathfrak{su}(2)$ fields leading to $\mathsf{T}_i(\boldsymbol{r}_n) = \exp\{\mathrm{i}\frac{q}{h}\sum_\alpha \boldsymbol{A}_\alpha(\boldsymbol{r}_n) \cdot \boldsymbol{u}_i \sigma^\alpha\} \in \mathrm{U}(2)$. As discussed below in more detail, in order to connect to Kitaev materials, we should allow for a more general tunneling than the unitary one allowed by background gauge fields, namely $\mathsf{T}_i \in \mathrm{GL}(2, \mathbb{C})$.

Let us now introduce the fermion-fermion interaction, which can be depicted by inter-layer Hubbard interactions of strength $U > 0$ in Fig. 2. Up to an trivial constant, this leads to the total 4-Fermi Hamiltonian

$$H = H_0 + \frac{U}{2}\sum_{\boldsymbol{n}} \left( \left(\psi_A^\dagger(\boldsymbol{r_n})\psi_A(\boldsymbol{r_n})\right)^2 + \left(\psi_B^\dagger(\boldsymbol{r_n})\psi_B(\boldsymbol{r_n})\right)^2 \right). \quad (3)$$

We refer to this model as the synthetic FHB, since one may think of the layer index $\ell = \mathrm{u}, \mathrm{d}$ as an internal degree of freedom of the fermions, making the implementation of the above Hubbard repulsion very natural. In this sense, the bilayer would be a synthetic geometry, and the inter-layer tunnelings (2) would define the connections along a synthetic 'dimension' [77, 78]. This perspective will become clearer when discussing a possible cold-atom implementation in Sec. IV.

## B. Strong-coupling Kitaev physics

In this subsection, we consider $|t_i^{\ell\ell'}| \ll U$, and present the effective strong-coupling model for the synthetic FHB at half filling. In this limit, the fermions tend to avoid the simultaneous population of the sites connected along the vertical AA stacking, as these have a large energy penalty $U$. Second-order processes can virtually populate these so-called doublons, leading to an effective spin-spin interaction that can be obtained applying a Schrieffer-Wolff transformation [79]. Coming back to the double-degenerate 'graphene' limit $t_i^{\mathrm{du}} = t_i^{\mathrm{ud}} = 0$, and $t_i^{\mathrm{uu}} = t_i^{\mathrm{dd}} = -t$, the effective kinetic exchange model would correspond exactly [79] to a Heisenberg model with anti-ferromagnetic nearest-neighbor couplings

$$H_{\mathrm{H}} = \sum_{\boldsymbol{n},i} J\,\boldsymbol{\sigma}_{\boldsymbol{r_n}} \cdot \boldsymbol{\sigma}_{\boldsymbol{r_n}+\boldsymbol{u}_i}, \quad J = \frac{t^2}{U}. \tag{4}$$

In the spin language $\boldsymbol{\sigma}_{\boldsymbol{r_n}} = (\sigma_{\boldsymbol{r_n}}^x, \sigma_{\boldsymbol{r_n}}^y, \sigma_{\boldsymbol{r_n}}^z)$, it is common to refer to the components of the spin operators as $x, y, z$, where

$$
\begin{aligned}
\sigma_{\boldsymbol{r_n}}^x &= a_{\boldsymbol{r_n},\mathrm{u}}^\dagger a_{\boldsymbol{r_n},\mathrm{d}} + a_{\boldsymbol{r_n},\mathrm{d}}^\dagger a_{\boldsymbol{r_n},\mathrm{u}}, \\
\sigma_{\boldsymbol{r_n}}^y &= \mathrm{i} a_{\boldsymbol{r_n},\mathrm{d}}^\dagger a_{\boldsymbol{r_n},\mathrm{u}} - \mathrm{i} a_{\boldsymbol{r_n},\mathrm{u}}^\dagger a_{\boldsymbol{r_n},\mathrm{d}}, \\
\sigma_{\boldsymbol{r_n}}^z &= a_{\boldsymbol{r_n},\mathrm{u}}^\dagger a_{\boldsymbol{r_n},\mathrm{u}} - a_{\boldsymbol{r_n},\mathrm{d}}^\dagger a_{\boldsymbol{r_n},\mathrm{d}},
\end{aligned} \tag{5}
$$

and similarly for $\boldsymbol{\sigma}_{\boldsymbol{r_n}+\boldsymbol{u}_i} = (\sigma_{\boldsymbol{r_n}+\boldsymbol{u}_i}^x, \sigma_{\boldsymbol{r_n}+\boldsymbol{u}_i}^y, \sigma_{\boldsymbol{r_n}+\boldsymbol{u}_i}^z)$, but exchanging the $a \leftrightarrow b$ fermionic operators of the $A \leftrightarrow B$ sublattices. In the above effective Hamiltonian, one assumes that there can only be one fermion per AA stack, such that the spin bilinears can be rewritten in terms of tensor products of the corresponding Pauli matrices, and one obtains a model of immobile interacting spins: a Heisenberg honeycomb model [20–22]. For the sake of completeness, we compare in Fig. 5 **(a)** the dynamics of a pair of interacting fermions in a Fermi-Hubbard dimer. Depending on the relative layer occupancies, the fermions can or cannot tunnel via the second-order process, which accurately matches that predicted flip-flop dynamics of the Heisenberg model.

The Heisenberg model has a continuous SU(2) symmetry regarding arbitrary rotations of the spins and, moreover, is isotropic with respect to any $C_3$ spatial rotation that swaps the nearest-neighbor bonds. In order to connect to Kitaev materials, we need to break explicitly both of these symmetries, which becomes possible when considering the more general bilayer tunneling matrices in Eq. (2). The additional flexibility in choosing the inter- and intra-layer tunnelings gives more freedom to design the effective spin-spin interactions. In particular, one finds a variety of paths for second-order tunneling processes where interference effects can be controlled to allow for the required bond anisotropies. Exploiting these interferences, we are able to find a spin model that supersedes Eq. (4) via a more general exchange tensor

$$H_{\mathrm{eff}} = \sum_{\boldsymbol{n},i} \left( \boldsymbol{\sigma}_{\boldsymbol{r_n}} \mathsf{J}_i \boldsymbol{\sigma}_{\boldsymbol{r_n}+\boldsymbol{u}_i} + \boldsymbol{D}_i \cdot (\boldsymbol{\sigma}_{\boldsymbol{r_n}} \wedge \boldsymbol{\sigma}_{\boldsymbol{r_n}+\boldsymbol{u}_i}) \right). \tag{6}$$

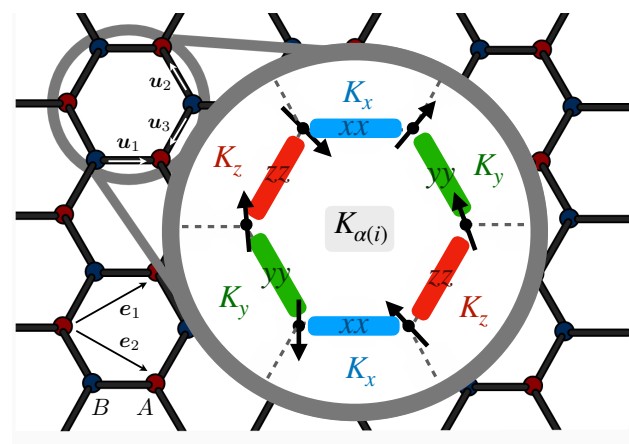

FIG. 3. **Strong-coupling Kitaev compass model:** Illustration of the different types of bond-dependent exchange interactions. The interplay fo the intra- and inter-layer tunnelings can lead to interference pathways that allow for control of the bond-exchange anisotropy, such that spins only interact though $xx$ interactions among the $\boldsymbol{u}_1$ nearest neighbors, $yy$ interactions among the $\boldsymbol{u}_2$ nearest neighbors, and $zz$ interactions among the $\boldsymbol{u}_3$ nearest neighbors.

Here, we have introduced the following couplings

$$
\mathsf{J}_i = \begin{pmatrix} J_i^x & \Gamma_i^z & \Gamma_i^y \\ \Gamma_i^z & J_i^y & \Gamma_i^x \\ \Gamma_i^y & \Gamma_i^x & J_i^z \end{pmatrix}, \quad \boldsymbol{D}_i = \left( D_i^z, D_i^y, D_i^z \right). \tag{7}
$$

In Fig. 4, we present a scheme of the main perturbations to Kitaev's bond-dependent exchange (see Fig. 3) that arise in Kitaev materials. In the left panel of Fig. 4, we see a standard Heisenberg-type exchange that does not distinguish any of the bonds, which would requires a more general exchange tensor with $J_1^x = J + K_x, J_1^y = J, J_1^z = J$, and the corresponding cyclic permutations for $\{J_2^\alpha, J_3^\alpha\}$. In the right panels, we depict the different off-diagonal symmetric exchange terms $\{\Gamma_i^\alpha\}$, and the correspondence to the $\Gamma, \Gamma'$ terms typically used in the context of Kitaev materials.

The elements in the diagonal of the matrix correspond in general to a so-called XYZ model [80, 81], which breaks the above SU(2) symmetry by involving spin anisotropies

$$
\begin{aligned}
J_i^x &= \frac{1}{U}\mathrm{Re}\left\{ t_i^{\mathrm{uu}} (t_i^{\mathrm{dd}})^* + t_i^{\mathrm{ud}} (t_i^{\mathrm{du}})^* \right\}, \\
J_i^y &= \frac{1}{U}\mathrm{Re}\left\{ t_i^{\mathrm{uu}} (t_i^{\mathrm{dd}})^* - t_i^{\mathrm{ud}} (t_i^{\mathrm{du}})^* \right\}, \\
J_i^z &= \frac{1}{2U}\mathrm{Re}\left\{ |t_i^{\mathrm{uu}}|^2 + |t_j^{\mathrm{dd}}|^2 - |t_i^{\mathrm{ud}}|^2 - |t_i^{\mathrm{du}}|^2 \right\}.
\end{aligned} \tag{8}
$$

Clearly, for the limit $t_i^{\mathrm{du}} = t_i^{\mathrm{ud}} = 0$, and $t_i^{\mathrm{uu}} = t_i^{\mathrm{dd}} = -t$, one recovers the antiferromagnetic Heisenberg model $J_i^x = J_i^y = J_i^z = t^2/U, \forall i \in \{1, 2, 3\}$. In addition, a Schrieffer-Wolff transformation also leas to spin-spin couplings involving different Pauli matrices according to a symmetric exchange pattern,

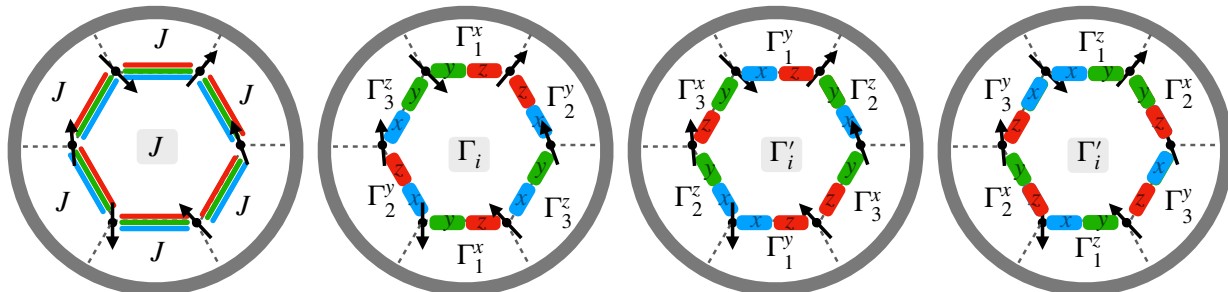

FIG. 4. **Kitaev's material corrections:** Illustration of representative kinetic exchange processes and the resulting spin-spin interactions on the honeycomb lattice. (Left panel) Standard Heisenberg-type exchange interaction arising from spin-independent tunneling, which leads to isotropic couplings that are identical on all bonds. The corresponding exchange tensor is characterized by $J_1^x = J + K_x$, $J_1^y = J$, $J_1^z = J$, with cyclic permutations for $\{J_2^\alpha, J_3^\alpha\}$. (Right panel) Symmetric off-diagonal exchange terms $\{\Gamma_i^\alpha\}$ that result from spin-dependent tunneling processes. These include both the $\Gamma$ and $\Gamma'$ interactions commonly used for extended Kitaev models in spin-orbit-coupled Mott insulators.

which is captured by the parameters

$$
\begin{aligned}
\Gamma_i^x &= \frac{1}{2U} \text{Im}\left\{ \left(t_i^{uu} + t_i^{dd}\right)\left(t_i^{ud} + t_i^{du}\right)^* \right\}, \\
\Gamma_i^y &= \frac{1}{2U} \text{Re}\left\{ \left(t_i^{uu} + t_i^{dd}\right)\left(t_i^{ud} - t_i^{du}\right)^* \right\}, \\
\Gamma_i^z &= \frac{1}{U} \text{Im}\left\{ t_i^{ud}\left(t_i^{du}\right)^* \right\}.
\end{aligned}
\tag{9}
$$

We note that these couplings vanish when there are no tunnelings connecting the two layers. Finally, there is also an anti-symmetric exchange in the effective spin model, formulated as a Dzyaloshinskii–Moriya term [82, 83] with couplings

$$
\begin{aligned}
D_i^x &= \frac{1}{2U} \text{Im}\left\{ \left(t_i^{uu} - t_i^{dd}\right)\left(t_i^{ud} - t_i^{du}\right)^* \right\}, \\
D_i^y &= \frac{1}{2U} \text{Re}\left\{ \left(t_i^{uu} - t_i^{dd}\right)\left(t_i^{ud} + t_i^{du}\right)^* \right\}, \\
D_i^z &= \frac{1}{U} \text{Im}\left\{ t_i^{uu}\left(t_i^{dd}\right)^* \right\}.
\end{aligned}
\tag{10}
$$

Note how these couplings also vanish in the absence of inter-layer tunnelings, provided that the intra-layer one is purely real, e.g. $t_i^{uu} = t_i^{dd} = -t \in \mathbb{R}$.

Let us now make contact with Kitaev's model [1] and, more generally, with Kitaev materials [16]. The Kitaev model requires a spatial bond-dependent anisotropy, such that $J_i^x = K_x \delta_{i,1}, J_i^y = K_y \delta_{i,2}, J_i^z = K_z \delta_{i,3}$ (see Fig. 3). When all other spin-spin terms vanish, one is left with

$$
H_K = \sum_n \left( K_x \sigma_{r_n}^x \sigma_{r_n+u_1}^x + K_y \sigma_{r_n}^y \sigma_{r_n+u_2}^y + K_z \sigma_{r_n}^z \sigma_{r_n+u_3}^z \right).
\tag{11}
$$

The direction of the interactions in spin space is thus correlated with the particular bond in real space, which is sometimes referred to as compass spin-spin interactions [84]. Here, we follow the nomenclature in Kitaev materials [16], labeling their corresponding strengths as $\{K_\alpha\}$ to recall that these are the *K*itaev bond-dependent couplings. As already mentioned, the kinetic exchange interactions in the equations above involve various second-order tunneling events that can destructively interfere. It is by exploiting this interference that we can

null the undesired contributions, achieving the desired model. For Kitaev's model, this can be achieved by setting

$$
\mathsf{T}_1 = \begin{pmatrix} t_1 & t_1 \\ -t_1 & -t_1 \end{pmatrix}, \;\; \mathsf{T}_2 = \begin{pmatrix} t_2 & it_2 \\ it_2 & -t_2 \end{pmatrix}, \;\; \mathsf{T}_3 = \begin{pmatrix} 0 & 0 \\ 2t_3 & 0 \end{pmatrix},
\tag{12}
$$

with $t_j \in \mathbb{R}$, such that the Kitaev model couplings read

$$
K_x = -\frac{2t_1^2}{U}, \quad K_y = -\frac{2t_2^2}{U}, \quad K_z = -\frac{2t_3^2}{U}.
\tag{13}
$$

One can readily check that $D_j^\alpha = \Gamma_j^\alpha = 0$, such that the effective strong-coupling limit of the half-filled FHB is a ferromagnetic Kitaev model. It is worth noting that most Kitaev material candidates are characterized by effective ferromagnetic Kitaev couplings, but these arise at a higher perturbative order that requires a Hund coupling $\mathcal{O}(t^2 J_H/U^2)$ [16]. As shown in Fig. 5 **(b)**, the dynamics developed by a pair of interacting fermions in a Fermi-Hubbard dimer accurately matches that predicted by the desired ferromagnetic Kitaev coupling, showing the same scaling $t^2/U$ as the standard Heisenberg couplings depicted in Fig. 5 **(a)** .

Our synthetic FHB allows us to obtain these ferromagnetic terms at the standard lower order of exchange couplings $\mathcal{O}(t^2/U)$ [79]. Moreover, we can also engineer an antiferromagnetic Kitaev model $K_\alpha \mapsto |K_\alpha|$ by considering

$$
\mathsf{T}_1 = \begin{pmatrix} t_1 & t_1 \\ t_1 & t_1 \end{pmatrix}, \;\; \mathsf{T}_2 = \begin{pmatrix} t_2 & -it_2 \\ it_2 & t_2 \end{pmatrix}, \;\; \mathsf{T}_3 = \begin{pmatrix} 2t_3 & 0 \\ 0 & 0 \end{pmatrix}.
\tag{14}
$$

Although the sign of the couplings may be irrelevant at the level of the exactly-solvable Kitaev model, the antiferromagnetic nature of the couplings can be relevant when other spin-spin terms (6) characteristic of Kitaev materials are also considered. In fact, this competition raises the possibility of a new emergent QSL only for the anti-ferromagnetic case at intermediate couplings and external magnetic fields [38, 39, 85].

An interesting feature of our synthetic FHB is that it allows for a simple design of additional spin-spin interactions, which are actually important to understand current experiments in the field of Kitaev materials. For instance, when considering the possible distortions of the crystal field mentioned

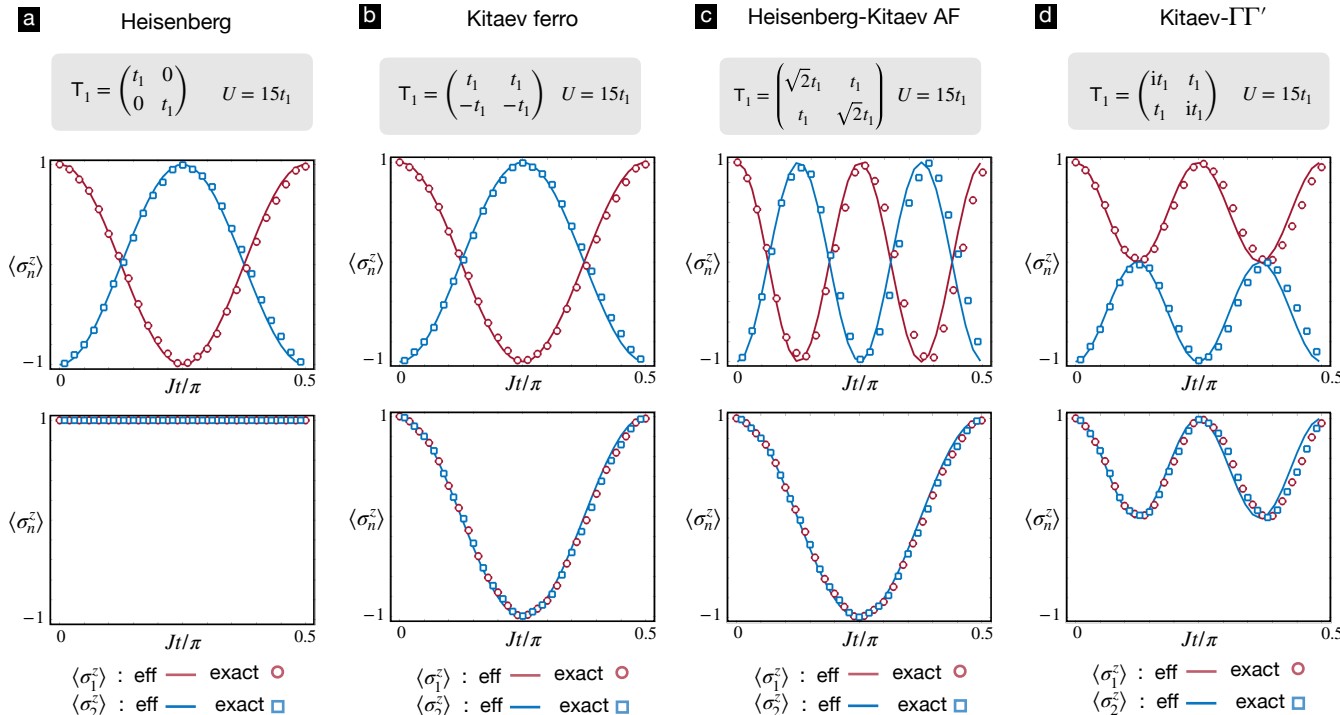

FIG. 5. **Numerical validation of tailored kinetic exchange:** We perform exact numerical simulations (circles) of a Hubbard bi-dimmer with various tunneling matrices $T_1$ and strong Hubbard interactions $U = 15t_1$, and compare to the effective exchange dynamics (solid lines) predicted by Eq. (8)-(9). **a** Standard Heisenberg exchange leading a flip-flop dynamics for $|\uparrow_1\downarrow_2\rangle \leftrightarrow |\downarrow_1\uparrow_2\rangle$ but no dynamics for parallel spins. **(b)** Kitaev $xx$ ferro coupling along $\boldsymbol{e}_1$, leading to a flip-flop of both $|\uparrow_1\downarrow_2\rangle \leftrightarrow |\downarrow_1\uparrow_2\rangle$ and $|\uparrow_1\uparrow_2\rangle \leftrightarrow |\downarrow_1\downarrow_2\rangle$. **(c)** Heisenberg-Kitaev coupling along $\boldsymbol{e}_1$, leading to a two-fold increase of the flip-flop dynamics of $|\uparrow_1\downarrow_2\rangle \leftrightarrow |\downarrow_1\uparrow_2\rangle$. **(d)** Kitaev materials coupling with competing bond-dependent and symmetric off-diagonal couplings, which leads to only partial exchange of both $|\uparrow_1\downarrow_2\rangle \leftrightarrow |\downarrow_1\uparrow_2\rangle$ and $|\uparrow_1\uparrow_2\rangle \leftrightarrow |\downarrow_1\downarrow_2\rangle$.

in the introduction, Kitaev materials also present competing Heisenberg interactions. In particular, the elongation or compression of these compounds changes the nature of the spin-orbit-coupled multiplets, turning them into standard spin orbitals, and introducing a standard Heisenberg interaction that becomes leading in the limit of large distortions [16]. In our synthetic FHB, one can explore the competition of Kitaev and Heisenberg interactions by setting the tunneling matrices to

$$T_1 = \begin{pmatrix} \tau_1 & t_1 \\ t_1 & \tau_1 \end{pmatrix}, \ T_2 = \begin{pmatrix} \tau_2 & -it_2 \\ it_2 & \tau_2 \end{pmatrix}, \ T_3 = \begin{pmatrix} \tau_3+t_3 & 0 \\ 0 & \tau_3-t_3 \end{pmatrix}, \tag{15}$$

where we have introduced $\tau_i = (t^2 + t_i^2)^{1/2}$ for each of the bonds $i \in \{1,2,3\}$. In this case, on top of the above antiferromagnetic Kitaev couplings, the diagonal elements of the spin exchange tensor (7) read

$$\begin{aligned} (J_1^x, J_1^y, J_1^z) &= \left(|K_x| + J, J, J\right), \\ (J_2^x, J_2^y, J_2^z) &= \left(J, |K_y| + J, J\right), \\ (J_3^x, J_3^y, J_3^z) &= \left(J, J, |K_z| + J\right), \end{aligned} \tag{16}$$

which have a bond-isotropic Heisenberg contribution (see left panel of Fig. 4). In spite of the modification of the bond-dependent tunnelings, one still finds $D_j^\alpha = \Gamma_j^\alpha = 0$, such that

the Hamiltonian is the so-called Heisenberg-Kitaev model

$$H_{\text{HK}} = \sum_{\boldsymbol{n},i} \left( J\,\boldsymbol{\sigma}_{\boldsymbol{r_n}} \cdot \boldsymbol{\sigma}_{\boldsymbol{r_n}+\boldsymbol{u}_i} + |K_{\alpha(i)}|\sigma_{\boldsymbol{r_n}}^{\alpha(i)}\sigma_{\boldsymbol{r_n}+\boldsymbol{u}_i}^{\alpha(i)} \right), \tag{17}$$

where we have introduced $\alpha(1) = x, \alpha(2) = y, \alpha(3) = z$ for the compass Kitaev interactions. We can thus easily interpolate between the purely Heisenberg and the purely Kitaev by tuning the ratio $t_i/t$, namely a Heisenberg $t \gg t_i$ and Kitaev $t \ll t_i$ limit. In Fig. 5 **(c)**, we consider again the Fermi-Hubbard dimer, but now adjust the tunnelings such that it leads to the desired Heisenberg-Kitaev couplings, finding again a clear agreement with the effective spin model.

Let us now move beyond, noting that the off-diagonal elements of the exchange matrix (7) also appear in the microscopic description of the Kitaev candidate materials. In fact, these terms arise as a consequence of crystal distortions, and have an order of magnitude that is similar to the Heisenberg term in the limit of small distortions [16]. In the context of Kitaev materials, contributions like $\Gamma_1^x$ and $\Gamma_1^y, \Gamma_1^z$ are typically referred to as $\Gamma_1$ and $\Gamma_1'$, respectively (see right panels of Fig. 4). The same nomenclature is used for other bonds with the respective cyclic permutations. In our synthetic FHB, by considering an arbitrary phase in the tunneling elements

$\tau_1 \to \tau_1 e^{\mp i\theta_1}, \tau_2 \to \tau_2 e^{\mp i\theta_2}$, we get access to these terms

$$\Gamma_1^x = \pm \frac{2t_1\tau_1}{U}\sin\theta_1, \;\; \Gamma_2^y = \pm\frac{2t_2\tau_2}{U}\sin\theta_2, \quad (18)$$

while all the remaining terms of the Kitaev-Heisenberg model (17) remain the same. The validity of this expression is confirmed in Fig. 5 **(d)** by comparing the exact dynamics of a Fermi-Hubbard dimer with the above effective couplings.

Let us close this subsection by noting that the effective Dzialosinkii-Moriya couplings (10) can also be switched by considering similar complex phases, but now allowing for layer-dependent angles. For instance, a simple way to obtain a Heisenberg XXZ antiferromagnet, $J_j^x = J_j^y = t^2/2U$, $J_j^z = t^2/U$, with a Dzialosinkii-Moriya coupling $D_j = -\sqrt{3}t^2/2U$, would be to use tunneling matrices $T_j = te^{-i\theta}(1+\sigma^z)/2 + te^{i\theta}(1-\sigma^z)/2$, with the phase $\theta = \pi/6$.

### C. Emergence of Kitaev-Mott insulators

As noted in the introduction, the synthetic FHB offers a simple playground to explore how the Kitaev QSLs, or other competing magnetic phases, arise from a non-interacting band theory as one gradually increases the Hubbard interactions.

In the non-interacting regime, the half-filled system is generally a semi-metal with excitations described by an effective Dirac QFT reminiscent of graphene [86, 87]. Moving to momentum space $\psi_{A,\mathbf{k}} = \sum_{\mathbf{n}} e^{i\mathbf{k}\cdot\mathbf{r_n}}\psi_A(\mathbf{r_n})/\sqrt{N_1 N_2}$, and likewise for the $B$ sub-lattice, one finds the following four-band model

$$H_0 = \sum_{\mathbf{k}\in\mathrm{BZ}_\bullet} \Psi_{\mathbf{k}}^\dagger \left( \sum_i \sigma^+ \otimes \mathsf{T}_i e^{i\mathbf{k}\cdot\mathbf{u}_i} + \mathrm{H.c.}\right)\Psi_{\mathbf{k}}, \quad (19)$$

where we have introduced the operator $\sigma^+ = \frac{1}{2}(\sigma^x + i\sigma^y)$ and the four-component spinor $\Psi_{\mathbf{k}} = (\psi_{A,\mathbf{k}}, \psi_{B,\mathbf{k}})^t$. We can now obtain the corresponding band structure by substituting the various tunnelings $\mathsf{T}_i$ discussed in the previous section, and diagonalizing this block off-diagonal matrix.

Considering the matrices in Eq. (14) and setting $t_1 = t_2 = t_3$ and $U = 0$, we obtain a simple four-band model with the energies displayed in the left panel of Fig. 6. In this case, the half-filled system consists of two filled bands, and one can see how the lowest occupied and highest unoccupied bands actually touch at two non-equivalent Fermi points at the corners of the Brillouin zone (BZ$_\bullet$) $\mathbf{K}_\mathrm{D}a = (0, 4\pi/3\sqrt{3})$ and $\mathbf{K}_\mathrm{D}'a = (2\pi/3, 2\pi/3\sqrt{3})$, as depicted in the contour plot on the left. These cone-like dispersions around $\mathbf{K}_\mathrm{D}, \mathbf{K}_\mathrm{D}'$ are reminiscent of graphene's band structure [86], and can be understood as an instance of fermion doubling in the lattice discretization of a Dirac quantum field theory (QFT) [88]. The groundstate of our synthetic FHB in the non-interacting and isotropic limit $t_1 = t_2 = t_3, U = 0$ is actually that of a Dirac semi-metal.

Inspired by the possibility of finding topological Lifshitz transitions in a single-species honeycomb lattice by varying the anisotropy [89, 90], we change one of the tunneling strengths introducing a bond-dependent anisotropy, e.g. $t_3$, the Dirac points $\mathbf{K}_\mathrm{D}, \mathbf{K}_\mathrm{D}'$ start approaching each other moving along one of the edges of the BZ (see the sequence of

contour plots following the red arrow on the left panel of Fig. 6 for various values of the ratio $t_3/t_1 = t_3/t_2$). Eventually, when the anisotropy reaches $t_3/t_1 = t_3/t_2 = 1/2$, the two Dirac points collide at the $M$ point of the BZ$_\bullet$, annihilating each other and opening a gap that effectively generates a mass for the Dirac fermions. This change of the Fermi surface can be interpreted as a topological phase transition, as each of the Dirac points carries a winding number of opposite sign $w_{\mathbf{K}_\mathrm{D}} = +1, w_{\mathbf{K}_\mathrm{D}'} = -1$ [91]. Their 'scattering' in momentum space thus results in an overall zero winding, resulting in a trivial band insulator. We note that similar effects have been found in the past for a honeycomb lattice under background non-Abelian SU(2) fields [76], although the cause and specific movement of the Dirac points is different in that case.

The more challenging question is to understand how, by gradually increasing the Hubbard repulsion $U > 0$ (green arrow in Fig. 6), the QSLs hosted by the Kitaev's honeycomb model (11) actually emerge, and what is the nature of the strongly-coupled critical points that separate this phase from the previous semi-metal or insulating phases. Following Kitaev's seminal work [1], the QSL phases in the regime of $U \gg t_i$ can be understood by introducing a parton construction in which each spin operator (5) results from the combination of a site ($c = c^\dagger$) and a bond ($d = d^\dagger$) Majorana fermion. These operators anti-commute with each other, and square to the identity, and can be used to represent the spins $\sigma_{\mathbf{r_n}}^\alpha = i d_{\mathbf{r_n}}^\alpha c_{\mathbf{r_n}}$ provided that the physical Hilbert space fulfills a parton constraint $d_{\mathbf{r_n}}^x d_{\mathbf{r_n}}^y d_{\mathbf{r_n}}^z c_{\mathbf{r_n}} |\psi\rangle = +|\psi\rangle$. Interestingly, in this formulation, the model reduces to a problem of Majorana fermions on the honeycomb lattice subjected to a background $\mathbb{Z}_2$ gauge field that lives on the links. This emerging gauge field is formed by a pair of bond Majoranas along the corresponding link $Z_{(\mathbf{r_n}, \mathbf{r_{n+u_i}})}^{\alpha(i)} = i d_{\mathbf{r_n}}^{\alpha(i)} d_{\mathbf{r_n}+\mathbf{u}_i}^{\alpha(i)}$, such that the bond-directional spin model in Eq. (11) can be exactly mapped onto a $\mathbb{Z}_2$ LGT of Majorana fermions

$$H_\mathrm{K} = \sum_{\mathbf{n},i} \frac{i}{2} K_{\alpha(i)} c_{\mathbf{r_n}} Z_{(\mathbf{r_n}, \mathbf{r_{n+u_i}})}^{\alpha(i)} c_{\mathbf{r_n}+\mathbf{u}_i} + \mathrm{H.c.}. \quad (20)$$

This LGT is not standard [14] in the sense that the gauge fields do not have any quantum dynamics of their own, and the matter is not a discretization of a Dirac field but a Majorana one, in which the dynamical $\mathbb{Z}_2$ charges are their own anti-charges. The $\mathbb{Z}_2$ gauge fields can be arranged in any possible $\pm 1$ classical configuration, setting a background for the Majorana fermions that affects the corresponding bandstructure. As argued by Kitaev, the groundstate lies in the 0-flux sector, and vortex excitations with a $\pi$-flux piercing an honeycomb plaquette come in pairs, are fully gapped and lead to a dispersionless band of so-called visons. On the contrary, the Majorana fermions residing at the sites describe the spinons, and form a dispersive band that can be easily obtained by diagonilizing the resulting 0-flux quadratic Majorana Hamiltonian after setting $Z_{(\mathbf{r_n}, \mathbf{r_{n+u_i}})}^{\alpha(i)} = +1$, $c_{B,\mathbf{k}} = \sum_{\mathbf{n}} e^{i\mathbf{k}\cdot\mathbf{r_n}}c(\mathbf{r_n})/\sqrt{N_1 N_2}$, and likewise for the $A$ sublattice. The free-fermion Hamiltonian for

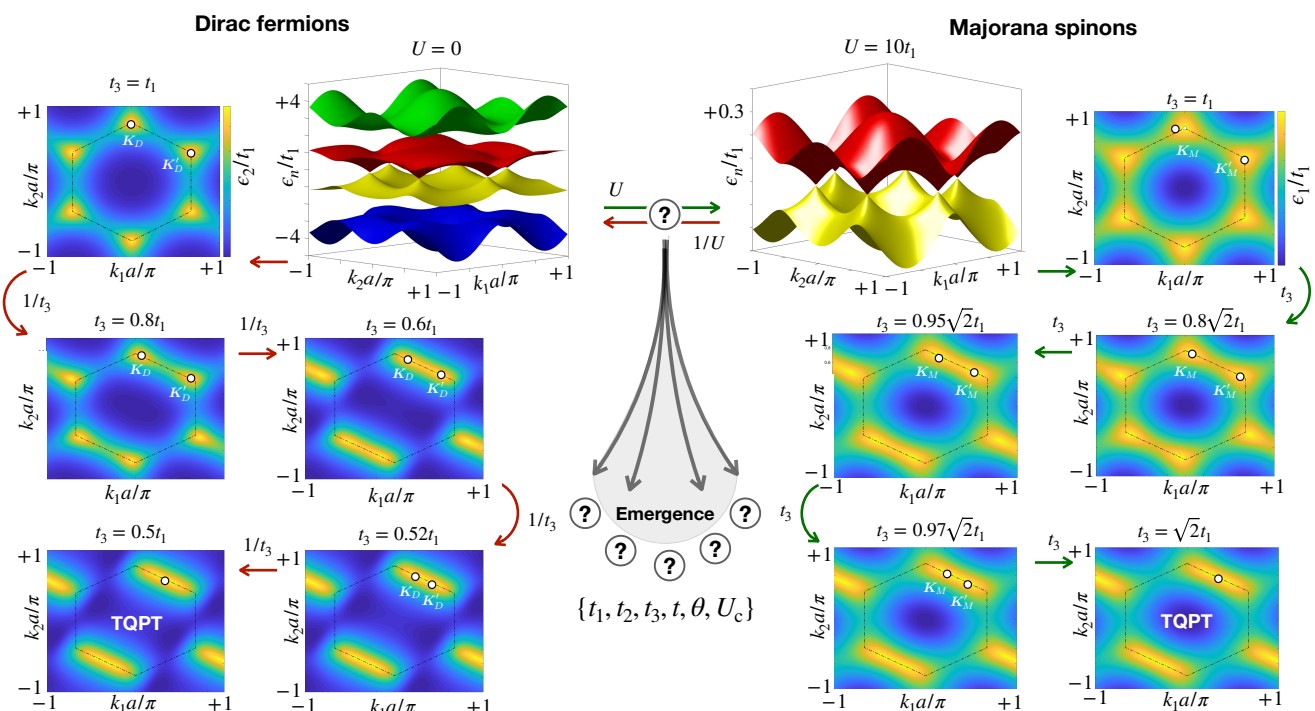

FIG. 6. **Emergence of Kitaev spin-liquid physics from the Hubbard bilayer:** (Left panel) Band structure of the non-interacting FHB model in the isotropic limit $t_1 = t_2 = t_3$ and $U = 0$, showing Dirac cones at two non-equivalent corners of the Brillouin zone, $\boldsymbol{K}_D$ and $\boldsymbol{K}_D'$. As a bond anisotropy is introduced by decreasing $t_3/t_1 = t_3/t_2$, the Dirac points approach each other and annihilate at $t_3/t_1 = 1/2$, opening a spectral gap and signaling a topological transition to a trivial band insulator. (Right panel) Majorana spinon band structure in the strong-coupling limit $U \gg t_i$, described by the effective Kitaev Hamiltonian. In the isotropic case $K_x = K_y = K_z$, two gapless Majorana cones appear at the same high-symmetry points $\boldsymbol{K}_M, \boldsymbol{K}_M'$ as in the Dirac case. Upon increasing $K_z/K_x = K_z/K_y = t_3^2/t_1^2$, the Majorana points move and eventually annihilate for $t_3 = \sqrt{2}t_1$, driving a transition to a gapped $\mathbb{Z}_2$ spin liquid phase connected to Kitaev's toric code.

this spinon band structure reads

$$H_K = \sum_{\boldsymbol{k} \in \mathrm{BZ}_\bullet} \left(c_{A,\boldsymbol{k}}^\dagger, c_{B,\boldsymbol{k}}^\dagger\right) \left(\sum_i \sigma^+ \tfrac{\mathrm{i}}{2} K_{\alpha(i)} \, e^{\mathrm{i}\boldsymbol{k}\cdot\boldsymbol{u}_i} + \mathrm{H.c.}\right) \begin{pmatrix} c_{A,\boldsymbol{k}} \\ c_{B,\boldsymbol{k}} \end{pmatrix}. \tag{21}$$

In analogy to the previous discussion of Dirac fermions in the four-band model arising from Eq. (19), we can again obtain the energies by substituting the specific Kitaev couplings (13) and diagonalising the corresponding matrix in momentum space. In contrast to the previous four-band model, we here obtain a two-band model with a band structure that again shows a pair of Fermi points in the fully isotropic limit $K_x = K_y = K_z$ (see the right panel of Fig. 6). The conic dispersion around these points, which again lie at the corners of the Brillouin zone $\boldsymbol{K}_M a = (0, 4\pi/3\sqrt{3})$ and $\boldsymbol{K}_M' a = (2\pi/3, 2\pi/3\sqrt{3})$, can now be understood in terms of a QFT of Majorana fields. Tuning the bond anisotropy leads to similar topological phase transitions as those described for the Dirac fermion case. As depicted by the contour plots along the green arrow of Fig. 6, as one increases the ratio of tunnelings $t_3/t_1 = t_3/t_2$ and, with it, the ratio of the Kitaev coupling strengths $K_z/K_x = t_3^2/t_1^2 = K_z/K_y$, the Majorana points start to move along one of the boundaries of the Brillouin zone.

In this case, when the tunneling reaches $t_3 = \sqrt{2}t_1$, the two Majorana points annihilate each other in a topological phase transition that generates a non-zero mass. In contrast with the Dirac fermion case, where the phase resulting from this process is a trivial band insulator, one finds that the annihilation of the Dirac points here leads to an exotic gapped QSL.

As argued by Kitaev, this phase connects to the deconfined phase toric code Hamiltonian [11], which requires setting $t_3 \gg t_1 + t_2$ in our case. Using perturbation theory to fourth order and a unitary rotation of the spin operators, we can find the following toric-code Hamiltonian

$$H_{\mathrm{TC}} = J_{\mathrm{eff}} \sum_\square \prod_{i \in \square} \tilde{\sigma}_i^z + J_{\mathrm{eff}} \sum_+ \prod_{i \in +} \tilde{\sigma}_i^x, \quad J_{\mathrm{eff}} = -\frac{t_1^4 t_2^4}{8U t_2^6}. \tag{22}$$

Here, the Pauli operators take into account the two possible ferromagnetic orderings in the leading $z$-bonds, namely $\tilde{\sigma}^x = |{\uparrow}{\uparrow}\rangle\langle{\downarrow}{\downarrow}| + |{\downarrow}{\downarrow}\rangle\langle{\uparrow}{\uparrow}|$, $\tilde{\sigma}^z = |{\uparrow}{\uparrow}\rangle\langle{\uparrow}{\uparrow}| - |{\downarrow}{\downarrow}\rangle\langle{\downarrow}{\downarrow}|$. This gapped QSL thus displays topological order and long-range entanglement, and forms the core to recent approaches to topological quantum codes for fault-tolerant quantum computation [12, 92]. From a many-body point of view, this gapped $\mathbb{Z}_2$ QSL is characterized by topological order: the absence of local order,

semionic excitations, degenerate ground states on homological non-trivial lattices, a non-vanishing topological entanglement entropy — all rooted in long-range entanglement.

Let us emphasise that studying the nature of the critical point $U_c$ that separates this QSL from the semi-metal, or otherwise insulating phases discussed previously, is an interesting open question. Moreover, the above arguments have only focused on the simplest cases where the limiting behaviors of our synthetic FHB are well known. In general, we can also include the additional parameters $t, \theta$ leading to additional spin-spin interactions (6), and also consider the case of antiferromagnetic Kitaev couplings (see the gray region in the central panel of Fig. 6). In this case, even the strongly-coupled limit is not known and a subject of considerable attention [16–18]. Understanding how the various possible QSLs and magnetically-ordered phases emerge and the nature of the critical strong-coupling points $\{U_c\}$ is thus a very interesting but complicated problem. The following section contains a detailed account for a quasi-one-dimensional version of this problem, which is amenable of very efficient matrix-product-state simulations. This reduction will allow us to address quantitatively some of the emergent features that would then connect to the strict QSL in the two-dimensional models. In the last section, we show how the full 2D problem could be addressed using cold-atom quantum simulators, thus using these devices to address problems that challenge our current knowledge and classical numerical capabilities.

## III. EMERGENCE OF KITAEV-TYPE PHYSICS ON A RIBBON

In this section, we provide a detailed quantitative account of the emergence of Kitaev physics when increasing the Hubbard repulsion $U > 0$. We focus on the extreme anisotropic limit $t_3 = 0$ for two reasons. On the one hand, the synthetic FHB breaks into a collection of disconnected inter-layer ribbons that can be seen as quasi-1D two-leg ladders. In this limit, the strongly-correlated properties of these ladders can be explored efficiently using matrix-product-state algorithms [33], allowing us to address quantitatively the emergence of Kitaev physics. On the other hand, by reducing to a ladder, it turns out that we can combine two leading themes in the topological quantum many-boy physics. As discussed in this section, the groundstate of the decoupled ladder at $t_3 = 0$ can be described by a band insulator that also has an effective QFT of massive Dirac fermions, albeit also displaying topological edge states that are localized to the boundaries of the ladder. This groundstate actually corresponds to a symmetry-protected topological (SPT) phase [93]. As one increases the Hubbard interactions beyond a certain critical point $U_c$, this SPT phase gives way to a precursor of the 2D Kitaev QSL, which can be understood as the groundstate of an alternating spin chain characterized by a non-local hidden order parameter instead of a local order parameter related to standard magnetism.

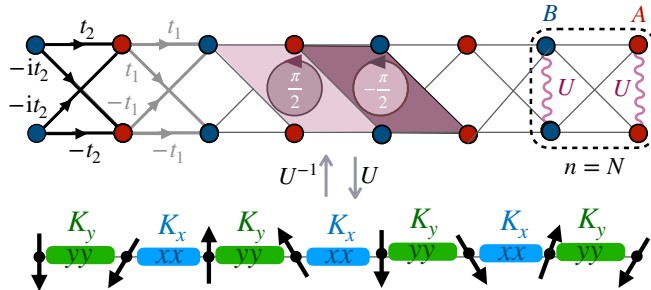

FIG. 7. **Fermi-Hubbard inter-layer ribbon:** Mapping of the synthetic Fermi-Hubbard model in the limit $t_3 = 0$, where the system decouples into independent inter-layer ribbons described by two-leg Hubbard dimerized ladders. The resulting ladder (upper panel) has a two-rung unit cell and retains the essential bond-dependent tunneling structure that allows for the emergence of Kitaev-like interactions in the strong-coupling regime (lower panel).

### A. Symmetry-protected topological ladder

In the $t_3 = 0$ limit, the connectivity of the resulting FHB reduces to a collection of decoupled inter-layer ribbons, which can be expressed as a two-leg ladder with a two-rung unit cell and inter-leg Hubbard interactions (see Fig. 7). This ladder model combines a cross-link structure and a finite flux piercing the rhombic plaquettes, with a dimerization leading to a 4-site unit cell. The former is reminiscent of the so-called Creutz ladder [94], while the enlarged dimerised pattern reminds of the Su-Schrieffer-Hegger model [95]. Various possible combinations of non-interacting Creutz-Su-Schrieffer-Hegger ladders have been discussed in detail in [96]. Our ladder model is similar in essence to the hourglass family of ladders explored in that work, but we have an extra dimerization of the flux.

Introducing the Nambu spinors $\psi_{A,n} = (a_{n,\mathrm{u}}, a_{n,\mathrm{d}})^{\mathrm{t}}$ and $\psi_{B,n} = (b_{n,\mathrm{u}}, b_{n,\mathrm{d}})^{\mathrm{t}}$, the Fermi-Hubbard ladder reads

$$H = \sum_n \left( \psi_{A,n+1}^{\dagger} \mathsf{T}_1 \psi_{B,n} + \psi_{A,n}^{\dagger} \mathsf{T}_2 \psi_{B,n} + \mathrm{H.c.} \right) + \frac{U}{2} \sum_{n,s} N_{s,n}^2, \tag{23}$$

where $s = A, B$, and $N_{A,n} = \psi_{A,n}^{\dagger} \psi_{A,n}, N_{B,n} = \psi_{B,n}^{\dagger} \psi_{B,n}$ are the fermion number in each rung of the $A, B$ sub-lattices, and we consider the tunneling matrices in Eq. (12) for $t_3 = 0$.

Let us follow the same steps used to illustrate the emergence of Kitaev materials in the 2D case. Switching off the Hubbard interactions $U = 0$, the model can again be diagonalized by using fermionic operators in momentum space $\psi_{s,k} = \sum_n \mathrm{e}^{ikan} \psi_{s,n}/\sqrt{N}$, where $k = -\frac{\pi}{a} + \frac{2\pi}{aN} j \in \mathrm{BZ}$ for $j \in \mathbb{Z}_N$. In analogy to Eq. (19) for the FHB, we find

$$H_0 = \sum_{k \in \mathrm{BZ}} \Psi_k^{\dagger} h_0(k) \Psi_k, \quad h_0(k) = \sigma^+ \otimes \left( \mathsf{T}_1 \mathrm{e}^{ika} + \mathsf{T}_2 \right) + \mathrm{H.c.}, \tag{24}$$

where we have introduced $\Psi_k = (\psi_{A,k}, \psi_{B,k})^{\mathrm{t}}$, a four-component spinor. The single-particle Hamiltonian $h_0(k)$ can then be readily diagonalized, leading to a four-band structure

$$\varepsilon_{\pm,b}(k) = \pm \varepsilon_b(k) = \pm \sqrt{f(k) + b\sqrt{g(k)}}, \tag{25}$$

where $f(k) = 2(t_1^2 + t_2^2 + t_1 t_2 \cos ka)$, $g(k) = f^2(k) - 4t_1^2 t_2^2$, and $b = \pm$. At half filling, the low-energy excitations occur around $K_D = 0$, involving the bands $\varepsilon_{+-}(k), \varepsilon_{-+}(k)$, which can be approximated by a Dirac-like dispersion

$$\varepsilon_{\pm\mp}(K_D + k') \approx \pm\sqrt{m^2 c^4 + c^2 k'^2}, \qquad (26)$$

where we have introduced an effective speed of light and mass

$$c^2 = t_1 t_2 a^2 \left( \frac{t_1^2 + t_2^2 + t_1 t_2}{|t_1 + t_2| \sqrt{t_1^2 + t_2^2}} - 1 \right), \qquad (27)$$

$$m^2 c^4 = 2\left(t_1^2 + t_2^2 + t_1 t_2\right) - 2|t_1 + t_2|\sqrt{t_1^2 + t_2^2}.$$

This analogy with the Dirac QFT can be formalized by projecting a long-wavelength expansion of Eq. (24) onto the two lowest-energy bands. We use the orthogonal projectors $P_k = |\varepsilon_{+-}(k)\rangle\langle\varepsilon_{+-}(k)| + |\varepsilon_{-+}(k)\rangle\langle\varepsilon_{-+}(k)|$ at each momenta, while $Q(k) = \mathbb{1} - P_k = |\varepsilon_{++}(k)\rangle\langle\varepsilon_{++}(k)| + |\varepsilon_{--}(k)\rangle\langle\varepsilon_{--}(k)|$ project onto the orthogonal complementary subspace. Performing a long-wavelength expansion around $k = K_D + k'$, with $|k'| \leq \Lambda_c \ll \pi/a$, we find

$$P_0 H P_0 \approx \int_{\Lambda_c} \frac{\mathrm{d}k'}{2\pi} \, \overline{\Psi}(k') \left( \gamma^1 c k' + mc^2 \right) \Psi(k'), \qquad (28)$$

such that $\overline{\Psi}(k') = \sqrt{a}\Psi_{k'}^\dagger \gamma^0, \Psi(k') = \sqrt{a}\Psi_{k'}$ can be identified with the adjoint and Dirac fields in momentum space. Here, we have introduced the projected gamma matrices

$$\begin{aligned}
\gamma^0 &= |\varepsilon_{+-}(0)\rangle\langle\varepsilon_{+-}(0)| - |\varepsilon_{-+}(0)\rangle\langle\varepsilon_{-+}(0)|, \\
\gamma^1 &= |\varepsilon_{+-}(0)\rangle\langle\varepsilon_{-+}(0)| - |\varepsilon_{-+}(0)\rangle\langle\varepsilon_{+-}(0)|,
\end{aligned} \qquad (29)$$

which fulfill the projected Clifford algebra $\{\gamma^\mu, \gamma^\nu\} = 2\eta^{\mu,\nu} P_0$, where $\eta = \mathrm{diag}(+1, -1)$ is the flat spacetime metric. It thus seems that the physics projected to the highest-occupied and lowest-unoccupied bands is a simple 1D version of the previous 2D Dirac fermions in the full FHB. In contrast to the 2D case, though, the mass is finite for any non-zero ratio of $t_2/t_1$, showing that the half-filled groundstate in this ladder corresponds to a gapped insulator rather than a semi-metal or an insulator depending on the ratio of the tunnelings. A subtler difference is that the 1D insulator actually corresponds to a symmetry-protected topological (SPT) phase.

The SPT nature of this phase becomes readily manifest by considering open boundary conditions. In this situation, instead of diagonalizing the Hamiltonian in momentum space, we consider directly the real-space model (23), momentarily setting $U = 0$. In Fig. 8, we represent the energy levels as a function of the tunneling ratio $t_2/t_1$. One can observe that the energy levels cluster into four energy bands corresponding to the bulk energies $\varepsilon_{b_1 b_2}$ in Eq. (25). In addition, there are a couple of persistent zero modes that correspond to the following orbitals localized at the boundaries of the ladder

$$|\varepsilon_\mathrm{L}\rangle = \frac{1}{\sqrt{2}}\left( ib_{1,\mathrm{u}}^\dagger + b_{1,\mathrm{d}}^\dagger \right)|0\rangle, \quad |\varepsilon_\mathrm{R}\rangle = \frac{1}{\sqrt{2}}\left( ia_{N,\mathrm{u}}^\dagger - a_{N,\mathrm{d}}^\dagger \right)|0\rangle. \qquad (30)$$

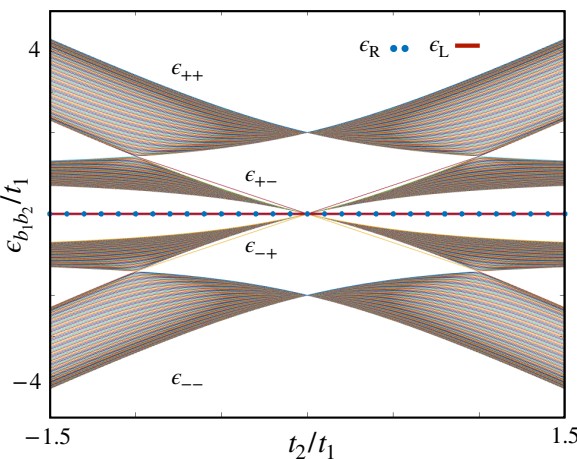

FIG. 8. **Energy spectrum of the non-interacting open ribbon**: For open boundary conditions, as we change the tunneling ratio $t_2/t_1$, the bulk energy bands group into four clusters $\varepsilon_{++}, \varepsilon_{+-}, \varepsilon_{-+}, \varepsilon_{--}$, corresponding to the analytic band structure derived from Eq. (25). In addition, two in-gap zero modes $\varepsilon_\mathrm{R}, \varepsilon_\mathrm{L}$ persist across a wide parameter range, corresponding to edge-localized orbitals described in Eq. (30). These are signatures of a symmetry-protected topological (SPT) phase protected here by inversion and sub-lattice symmetries.

The bulk manifestation of this SPT phase can be studied by computing the Berry [97] or Zak's phase [98]

$$\varphi_{\pm,b} = \int_\mathrm{BZ} \mathrm{d}k \langle \varepsilon_{\pm,b}(k) | \mathrm{i}\partial_k | \varepsilon_{\pm,b}(k) \rangle, \qquad (31)$$

associated to the negative-energy bands filled in the groundstate. The non-trivial SPT phase is signaled by $\varpi := \varphi_{-,-} + \varphi_{-,+} = \pi$. This Berry phase can be alternatively expressed as $\varphi = i \int_0^{2\pi} \mathrm{d}\theta \, \langle \psi(\theta) | \partial_\theta \psi(\theta) \rangle$, where $|\psi(\theta)\rangle$ is the ground state of the Hamiltonian with a single bond twisted as $t \to te^{i\theta}$ [99]. In Fig. 11(a), we present the results for the ground-state Berry phase $\varphi$ calculated using a MPS-based density matrix renormalization group (DMRG) algorithm [100]. At $U = 0$, we show how the Berry phase is indeed quantized to $\varphi = \pi$ in the whole parameter space $t_1/t_2$, in agreement with the analytical results discussed above. In the next section, we will show how these results are modified as we turn on the Hubbard interactions.

So far, we have not discussed the global protecting symmetries of this topological phase. We find that time-reversal symmetry $\mathscr{T}$ acts on the single-particle Hamiltonian as $T^\dagger h_0^*(-k)T = h_0(k)$, where $T = \sigma^z \otimes \sigma^x$. In addition, charge-conjugation $\mathscr{C}$ requires $C^\dagger h_0^*(-k)C = -h_0(k)$, which is achieved by $C = \mathbb{1}_2 \otimes \sigma^x$. The sub-lattice symmetry $\mathscr{S} = \mathscr{T} \circ \mathscr{C}$, sometimes also called chiral symmetry in the context of SPT phases, amounts to $S^\dagger h_0^*(k)S = -h_0(k)$ with $S = TC = \sigma^z \otimes \mathbb{1}_2$. According to the classification of SPT phases, the quantiszd $\pi$ value of the Berry phase of Fig. 11(a) and the edge states of Fig. 8 are the bulk-boundary manifestations of a SPT phase in class BDI.

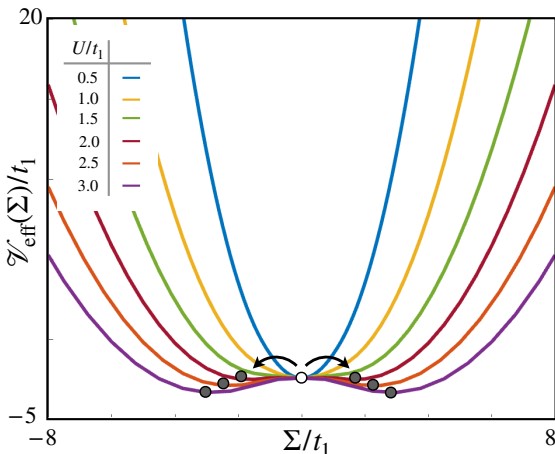

FIG. 9. **Large-$N_f$ effective potential symmetry beraking:** We represent $\mathscr{V}_{\text{eff}}(\Sigma)$ for the auxiliary scalar field $\Sigma$. As the Hubbard repulsion $U$ increases, the effective potential transitions from a single-well to a double-well structure, indicating spontaneous symmetry breaking and the emergence of a non-zero scalar condensate signaling the extinction of the SPT phase.

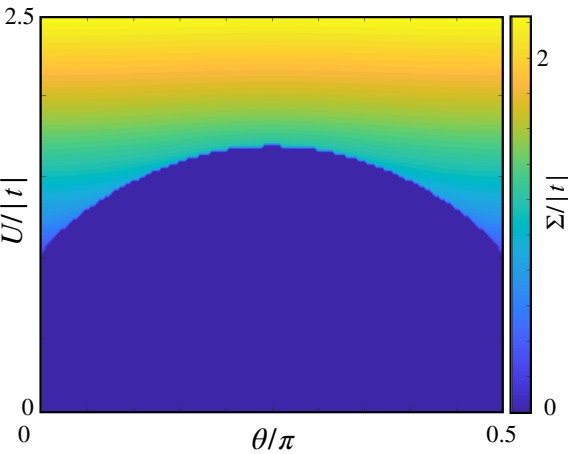

FIG. 10. **Large-$N_f$ phase diagram:** contour plot of the scalar condensate $\Sigma$ as a function of interaction strength $U/|t|$ and tunneling anisotropy, parametrized by $t_1 = |t|\cos\theta$, $t_2 = |t|\sin\theta$. The blue region corresponds to the SPT phase characterized by a vanishing condensate, which gives way to a correlated phase with broken $\mathscr{T}, \mathscr{C}$, and $\mathscr{S}$ symmetries beyond a critical interaction strength.

### B. Mott insulators and the Kitaev spin chain

We are now interested in switching on the repulsion $U > 0$, and quantifying its effect on the above SPT phase. We will proceed by using a path-integral formalism [101], and introducing an auxiliary scalar field $\sigma(x)$ via a Hubbard-Stratonovich transformation to deal with the quartic density-density interactions (23). In this approach, one works with the partition function, and can derive non-perturbative analytical predictions by using a large-$N_f$ expansion that is analogous to a Hartree-Fock approximation. We formally introduce $N_f$ fermion flavors $\{\psi_{A,n,f}, \psi_{B,n,f}\}_{f=1}^{N_f}$, each of which is a copy of the BDI insulator, and couple them via the Hubbard interactions (23). To allow for a well-defined $N_f \to \infty$ limit, we need to rescale the interaction strength to

$$V_{\text{int}} = \frac{U}{2N_f} \sum_{n=1}^{N} \left( \left( \sum_{f=1}^{N_f} \psi_{A,n,f}^\dagger \psi_{A,n,f} \right)^2 + \left( \sum_{f=1}^{N_f} \psi_{B,n,f}^\dagger \psi_{B,n,f} \right)^2 \right). \tag{32}$$

The Hubbard-Stratonovich transformation allows to rewrite this quartic term by a local coupling between fermion bilinears and the scalar auxiliary fields. One can then proceed by calculating the effective action arising from the leading Feynman diagrams at large-$N_f$, which correspond to those with a single fermion loop connected to zero-momentum legs of the auxiliary scalar field [102]. Although the scalar field can in principle have any spatial dependence, it is customary in half-filled problems to assume that it will condenses to a possibly non-zero value in the large-$N_f$ limit, inducing a so-called scalar condensate $\Sigma$. This leads to the following effective potential

$V_{\text{eff}}(\Sigma) = N_f N \mathscr{V}_{\text{eff}}(\Sigma)$ per site and per flavor

$$\mathscr{V}_{\text{eff}}(\Sigma) = \frac{1}{U}\Sigma^2 - \frac{1}{N}\sum_{k\in\text{BZ}}\sum_{b=\pm}\int_{\mathbb{R}}\frac{d\omega}{2\pi}\log\Big(\omega^2 + \varepsilon_b^2(k,\Sigma)\Big), \tag{33}$$

where we integrate over all possible Matsubara frequencies $\omega \in \mathbb{R}$ to recover a zero-temperature prediction [101]. Here, the energy bands $\varepsilon_b(k,\Sigma)$ resulting from the scalar condensate can still be expressed by Eq. (25), after making the substitutions $f(k) \mapsto f(k,\Sigma) = 2(t_1^2 + t_2^2 + t_1 t_2 \cos ka) + (\Sigma/2)^2$, $g(k) \mapsto g(k,\Sigma) = f^2(k,\Sigma) - (4t_1^2 t_2^2 + (\Sigma/2)^4 - t_1 t_2 \Sigma^2 \cos(ka))$. In Fig. 9, we represent various instances of this effective potential for $t_1 = t_2$. As the Hubbard repulsion increases, the effective potential changes from a single to a double well, signaling the formation of a non-zero vacuum expectation value $\pm\Sigma_0$ corresponding to a condensate of particles and holes. This transition is associated to a spontaneous breakdown of the $\mathscr{T}, \mathscr{C}$ and $\mathscr{S}$ symmetries introduced below Eq. (30).

At this large-$N_f$ limit, a non-zero condensate translates into a Hartree-Fock-type single-particle Hamiltonian

$$h_0(k) \mapsto h_{\text{eff}}(k,\Sigma) = h_0(k) \pm \Sigma_0 \mathbb{1}_2 \otimes \sigma^z, \tag{34}$$

which clearly breaks all of the above global symmetries. Hence, the single- to double-well transition of the effective potential in Fig. 9 also signals the breakdown of the protecting symmetry of the SPT phase, which gives way to a different symmetry-broken groundstate. This is depicted in Fig. 10, which shows how the scalar condensate can attain a non-zero value for sufficiently-large interactions $U/|t|$, where we parametrize the tunneling anisotropy with an angle $t_1 = |t|\cos\theta$, $t_2 = |t|\sin\theta$ for $\theta \in [0, \pi/2]$. In this figure, the blue area represents a large-$N_f$ SPT phase that is adiabatically connected to the BDI phase of Fig. 8. Depending on

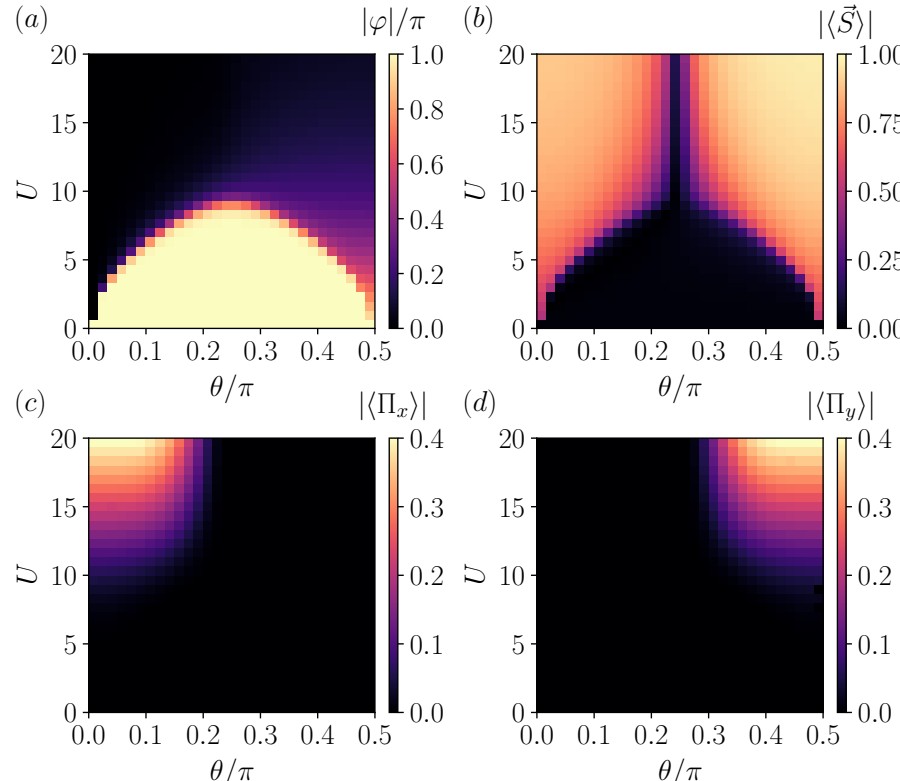

FIG. 11. **DMRG phase diagram:** (a) and (b) show the Berry phase $\varphi$ and the total magnetization $\langle \vec{S} \rangle$, respectively, as a function of the angle $\theta$ and the Hubbard interactions $U$. In the SPT phase, $\varphi$ is quantized to one and $\langle \vec{S} \rangle$ vanishes. For large enough values of $U$, a different phase emerges where $\varphi$ drops to zero, while $\langle \vec{S} \rangle$ acquires a non-zero value in two different regions and vanish at $\theta = \pi/4$. The results are obtained for a system of size $L = 50$ and open boundary conditions. (c) and (d) show the value of the string order parameters $\langle \Pi_x \rangle$ and $\langle \Pi_x \rangle$, respectively, as a function of $\theta$ and $U$, indicating how string order emerges at large enough values of $U$. The results are obtained for a system of size $L = 50$ and periodic boundary conditions. In both cases, we calculated the ground state using DMRG with a MPS of bond dimension $D = 300$.

the ratio of the tunnelings $t_2/t_1$, the large-$N_{\rm f}$ SPT phase manifests a higher or lower robustness to the interaction. In any case, it will eventually give way to a phase characterized by a symmetry-broken scalar condensate.

The large-$N_{\rm f}$ analysis performed above allowed us to predict that the SPT phase disappears in favor of a scalar condensate. We expect that the critical line that encloses the SPT phase will certainly move when considering the $N_1 = 1$ limit of relevance to our FHB. What might be more surprising is the fact that this large-$N$ prediction misses the nature of the resulting phase, which cannot be fully characterized by a broken symmetry and a local order parameter $\Sigma_0$. In retrospect, this is actually not that surprising, given that the large-$N_{\rm f}$ formalism selects a specific condensation channel when rewriting the interactions, and is in essence a mean-field theory that cannot capture the subtleties of quantum disordered phases such as QSLs. Even if the 1D model is not strictly speaking the Kiatev's QSL, we now show that it might be understood as the precursor to them, as it can actually display non-trivial properties such as non-local string-order parameters.

We now employ DMRG to *(i)* predict the correct phase boundary surrounding the SPT phase, and *(ii)* explore the new phase that appears for strong Hubbard repulsion. In

Fig. 11**(a)**, we depict the phase diagram using the many-body Berry phase, which is calculated by twisting a bond of the many-body groundstate as discussed in the previous section. The contour plot of the Berry phase serves to identify the lobe structure of the correlated SPT phase, showing qualitative agreement with the large-$N_{\rm f}$ prediction of Fig. 10. Moreover, in Fig. 11**(b)** we show how, while the total magnetization $\langle \vec{S} \rangle$ vanishes in the SPT phase, it develops a non-zero value for larger values of the Hubbard repulsion $U$. Although this might be consistent with the scalar condensate $\Sigma_0$ predicted by the large-$N_{\rm f}$, there are crucial differences.

To uncover these differences, we compute with DMRG the following string-order-like parameters

$$\Pi_x = \prod_n \sigma_n^x, \quad \Pi_y = \prod_n \sigma_n^y. \tag{35}$$

which are inspired by the physics of the 1D Kitaev model [103, 104]. We know that in the limit $t_1, t_2 \ll U$, the kinetic exchange of the Hubbard ladder will yield the corresponding 1D chain partition of the full Kitaev model

$$H = \sum_n (K_x \sigma_{2n}^x \sigma_{2n+1}^x + K_y \sigma_{2n-1}^y \sigma_{2n}^y), \tag{36}$$

a spin model with interspersed $\sigma^x\sigma^x$ and $\sigma^y\sigma^y$ interactions of strength $K_x = -2t_1^2/U$ and $K_y = -2t_2^2/U$ (see Fig. 7). Even in this model has two non-commuting terms that compete against each other, the groundstate turns out to be a either a valence-bond solid $|\psi_0\rangle = \prod_n |\psi_{2n,2n-1}^x\rangle$ with Bell states on the even bonds $|\psi^x\rangle = (|\uparrow\uparrow\rangle + |\downarrow\downarrow\rangle)/\sqrt{2}$ for $K_x \gg K_y$, or a valence-bond solid $|\psi_0\rangle = \prod_n |\psi_{2n-1,2n}^y\rangle$ with Bell states on the odd bonds $|\psi^y\rangle = (|\uparrow\uparrow\rangle - |\downarrow\downarrow\rangle)/\sqrt{2}$ for $K_y \gg K_x$. For arbitrary values of the ratio, the 1D Kitaev model can be mapped onto an Ising model in a transverse field by a duality transformation [103], such that one finds $\Pi_x = (1 - K_y^2/K_x^2)^{1/4}$ or $\Pi_y = (1 - K_x^2/K_y^2)^{1/4}$ in the $N \to \infty$ limit. Figures 11(c) and (d) show how these two different hidden non-local orders emerge as we depart from the SPT phase, and tend to the expected unit values for sufficiently-large repulsion $U$. The $\Pi_x$ and $\Pi_y$ orders are separated by a transition at $t_1 = t_2$, which also agrees with the critical point of the 1D Kitaev model $K_x = K_y$.

Even if this hidden string order resonates with the properties of QSLs, we note that the above valence-bond-type groundstates are short-range entangled. Moreover, by introducing a simple energy imbalance between the legs of the ladder, one can show that they connect adiabatically to the standard disordered paramagnet. However, this phase can be considered as precursor of Kitaev's QSL since, in the process of inter-connecting gradually more and more ribbons, one would map to the Kitaev ladders studied in [103] that develop the full phase diagram of the Kitaev honeycomb model. In fact, already for a pair of ribbons, and this a Kitaev two-leg ladder, the critical lines identified by these hidden string-order parameters already delimit the exact locations in parameter space of the gapped $\mathbb{Z}_2$ QSL. As one increases the number of ribbons further, the additional critical lines completely fill an intermediate region that corresponds to the Kitaev's gapless QSL phase. In future works, it would be interesting to extend our DMRG studies of the itinerant fermion models to these coupled ribbons to see how the QSL emerges gradually, and how the 2D Mott insulating regime is consistent with Kitaev's construction.

## IV. ULTRACOLD FERMIONS IN RAMAN LATTICES

In this section, we will show how the FHB Hamiltonian (3) introduced above can be realized in a system of ultracold atoms trapped in optical lattices. These are generated by counter-propagating laser beams and, in the limit of deep optical potentials, the second-quantized atomic Hamiltonian can be described by a tight-binding Fermi-Hubbard (FH) model [105–108], namely

$$H_{\rm FH} = -t \sum_{\boldsymbol{n},j,\sigma} \left( c_{\boldsymbol{n},\sigma}^\dagger c_{\boldsymbol{n}+\mathbf{e}_j,\sigma} + {\rm H.c.} \right) + U \sum_{\boldsymbol{n}} n_{\boldsymbol{n},\uparrow} n_{\boldsymbol{n},\downarrow}. \quad (37)$$

Here $c_{\boldsymbol{n},\sigma}^\dagger$ ($c_{\boldsymbol{n},\sigma}$) denote the creation (annihilation) fermionic operator at lattice site $\boldsymbol{n}$ and internal atomic state $\sigma \in \{\downarrow,\uparrow\}$, and $n_{\boldsymbol{n},\sigma} = c_{\boldsymbol{n},\sigma}^\dagger c_{\boldsymbol{n},\sigma}$ is the corresponding atomic occupation

number. Finally, $t$ corresponds to the nearest-neighbor tunneling coupling between sites $\boldsymbol{n}$ and $\boldsymbol{n}+\mathbf{e}_j$ along the direction $j \in \{x,y\}$, and $U$ is the on-site Hubbard interaction strength.

We will now generalize this construction to obtain the FHB (3). The first step is to map the honeycomb lattice into a brick-wall lattice, such that the $\boldsymbol{u}_1$ bonds correspond to the $x$ direction, and the $\boldsymbol{u}_2$ and $\boldsymbol{u}_3$ bonds are aligned in the $y$ direction [Fig. 12(a)]. A brickwall optical lattice can be implemented using three retro-reflected laser beams, two in the $x$ axis (with different frequencies) and one in the $y$ axis, linearly polarized in $z$ [Fig. 12(b)], and with the same wavelength $\lambda$ [109, 110]. This setup gives rise to the following optical potential,

$$
\begin{aligned}
V(x,y) = &-V_{x,1}\cos^2(kx) - V_{x,2}\cos^2(kx+\theta/2) \\
&- V_y\cos^2(ky) - 2\alpha\sqrt{V_{x,1}V_y}\cos(kx)\cos(ky),
\end{aligned}
\quad (38)
$$

where $V_{x,1}$, $V_{x,2}$ and $V_y$ are the lattice depths corresponding to each beam, $k = 2\pi/\lambda$, and $\alpha$ is the visibility of the interference pattern. The brickwall structure is then obtained by choosing the laser intensities such that $V_{x,1}/E_{\rm R} = 0.28$, $V_{x,2}/E_{\rm R} = 4.0$ and $V_{x,1}/E_{\rm R} = 1.8$, and the relative phase $\theta = \pi$ [109]. Moreover, by tuning $\theta$ away from $\pi$ by an angle $\varepsilon$ one can create a staggered energy shift $\Delta_\varepsilon$ at the potential minima corresponding to each sub-lattice. We combine this staggered potential with a gradient in both the $x$ and $y$ axes, obtained by implementing the corresponding lattice acceleration [111] or a magnetic-field gradient [112]. We choose this gradient to be $\Delta = \Delta_\varepsilon$, giving rise to the bond structure depicted in Fig. 12(c).

We use this potential to trap two-component fermionic atoms, and we identify two internal hyperfine levels $\uparrow/\downarrow$ with the $u/d$ states in the FHB. The system is thus described by a FH Hamiltonian (37) on an honeycomb lattice, where only intra-layer tunneling amplitudes have non-zero values. We now show how to generate the remaining interlayer tunneling processes. Contrary to previous proposals, we will employ Raman lattices [67–70] to generate the corresponding tunneling matrices (2), avoiding in this way the residual photon scattering associated to fermionic atoms in spin-dependent optical potentials [52], or the possible heating that may appear in Floquet-driven systems [113]. In the following, we will start by focusing on the ferromagnetic case given by Eq. (12).

Following Refs. [114–116], we consider additional standing waves together with Raman beams along the $x$ and $y$ axes [Fig. 12(b)]. Specifically, the inter-layer tunnelings in the $x$ ($y$) axis are assisted by Raman beams along the $y$ ($x$) axis polarized in the $x$ ($z$) direction [Fig. 12(a)]. Due to the difference in polarization, the Raman beam together with the $z$ ($x$)-polarized standing wave along the $x$ ($y$) axis give rise to two-photon spin-changing Raman processes [Fig. 12(c)]. Moreover, the different spatial periodicity with respect to the original brickwall lattice, one can realize that on-site spin-flip terms vanish while spin-changing tunneling processes have a non-zero contribution. In the tight-binding limit, these tunnelings processes take the following form,

$$H_{{\rm R},j} = -\sum_{\boldsymbol{n}} \left[ \mathrm{i}\tilde{t}_j \mathrm{e}^{\mathrm{i}(\delta_j t - \phi_{\boldsymbol{n},j})} \left( c_{\boldsymbol{n},\downarrow}^\dagger c_{\boldsymbol{n}+\mathbf{e}_j,\uparrow} - c_{\boldsymbol{n},\downarrow}^\dagger c_{\boldsymbol{n}-\mathbf{e}_j,\uparrow} \right) + {\rm H.c.} \right],$$

$$(39)$$

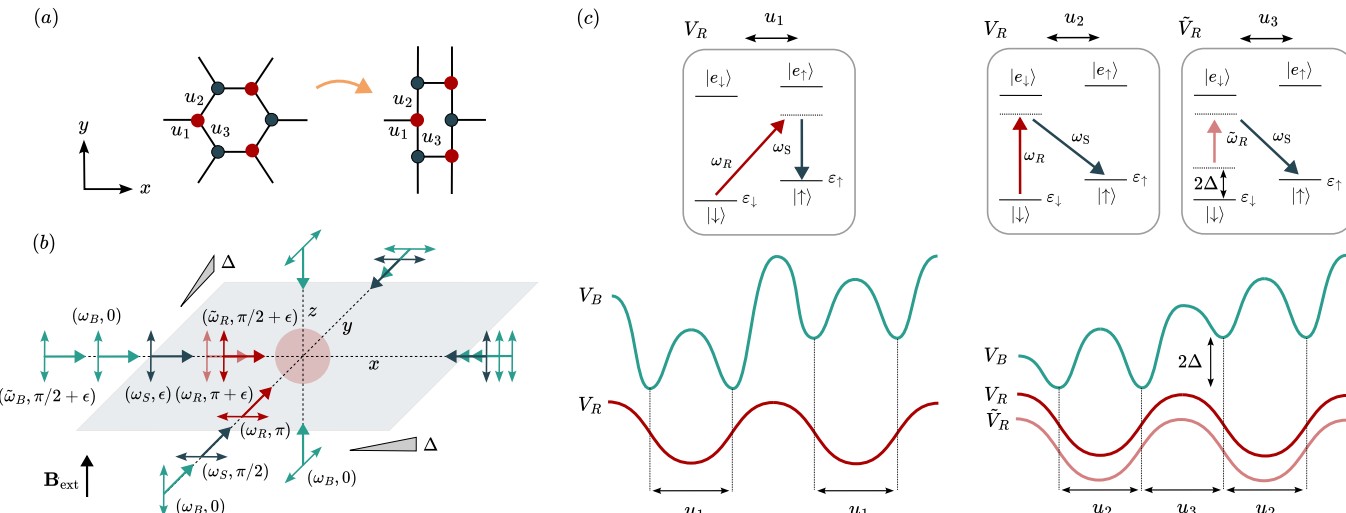

FIG. 12. **Fermionic atoms in a Raman honeycomb lattice:** (a) We simulate an honeycomb lattice by constructing an equivalent brickwall optical lattice. (b) This can be achieved by using two interfering retro-reflected laser beams in the $x$ and $y$ direction, together with an extra beam in the $x$ direction with a relative phase of $\pi/2 + \epsilon$, where $\epsilon > 0$ introduces a staggered chemical potential $\Delta_\epsilon$ (see main text). An additional standing wave in the $z$ direction confine the atoms to a 2D plane. In the figure, the lasers used for the brickwall lattice are depicted in green, and the corresponding arrows indicate the direction of propagation and polarization. On top of this lattice, Raman potentials $V_R$ and $\tilde{V}_R$ are formed by interfering additional standing waves (blue) together with two Raman beams in the $x$ direction, and one in the $y$ direction (red). All the required frequencies and phases $(\omega, \varphi)$ are indicated in the figure. Finally, a magnetic field $\mathbf{B}_{\text{ext}}$ sets the spin quantization to the $z$ direction, and a gradient $\Delta$ is imposed in both the $x$ and $y$ directions using either a magnetic field gradient or through lattice acceleration. (c) In the $x$ direction, the Raman potential $V_R$ gives rise to spin-flipping tunneling terms, obtained from two-photon processes resulting from the interference of a one standing and one Raman beam. The brickwall structure only allows tunneling processes for every second bond, as required for the $u_1$ bonds in the original honeycomb lattice (a). For $\Delta_\epsilon = \Delta$, even bonds in the $y$ direction have an associated energy imbalance of $2\Delta$, and the imbalance is zero for the odd ones. As a result, two different Raman beams can generate the spin-flipping tunneling elements for the bonds $u_2$ and $u_3$ of the honeycomb lattice, provided that the corresponding resonant conditions are fulfilled (insets), allowing to tuned their phases independently.

where $\tilde{t}_j$ is the Raman-assisted tunneling coupling along the $j$ axis, and we define $\phi_{\boldsymbol{n},j} = \phi_j - \pi(n_x + n_y)$, where $\phi_j$ is the relative phase between the standing wave and the Raman beam. Finally, $\delta_j = \omega_S - \omega_{R,j} - (\varepsilon_\downarrow - \varepsilon_\uparrow)$ is the corresponding detuning for the two-photon Raman transition, with $\omega_S$ and $\omega_{R,j}$ the frequencies of the standing wave and the Raman beams, respectively, and $\varepsilon_\sigma$ is the electronic energy for the level $\sigma$, controlled by the external magnetic field $\mathbf{B}_{\text{ext}}$ [see Fig. 12(c)].

Due to the two-site unit cell imposed by the staggered onsite potential and the linear gradients described above [see Fig. 12(b)], the tunneling matrices corresponding to the three bonds of the original honeycomb lattice can be tuned independently using three different Raman beams. In particular, in the $y$ axis we can choose one Raman beam with $\delta_y = 0$ and another one with $\delta_y = 2\Delta$, such that they only become on-resonant for the $\boldsymbol{u}_2$ and $\boldsymbol{u}_3$ bonds, respectively [see Fig. 12(c)]. Finally, after applying the gauge transformation $c_{\boldsymbol{n},\uparrow} \to e^{i\pi(n_x + n_y)} c_{\boldsymbol{n},\uparrow}$, the tunneling matrices corresponding to the ferromagnetic Kitaev model are recovered by choosing the Raman phases appropriately, as indicated in Fig. 12(b). We note that the tunneling couplings along the three different bonds of the honeycomb lattice can be tuned independently, allowing e.g. to drive the system along the abelian to non-abelian topologically ordered phases.

Using similar constructions, the antiferromagnetic Kitaev model can be also implemented, as well as the different perturbations present in Kitaev materials discussed in Sec. II. For instance, the tunneling matrices (15) associated to the Heisenberg-Kitaev model (17) could be implemented by using a modulated magnetic field to generate spin-dependent tunneling elements along the $\mathbf{u}_3$ bonds [117]. Moreover, tuning the relative tunneling coupling between intra- and inter-layer processes can be done in a straightforward manner by modifying the intensity of the Raman beams with respect to the standing waves. Finally, we note that the implementation of the synthetic FHB introduced in this work is not restricted to the Raman lattice setup discussed here, and could be thus improved by future experimental developments.

## V. CONCLUSIONS AND OUTLOOK

This work has examined a synthetic Fermi-Hubbard bilayer (FHB) model that supports the emergence of bond-directional spin interactions similar to those that appear in Kitaev's celebrated honeycomb model [1]. Considering a bilayer honeycomb lattice with complex tunneling and inter-layer Hubbard interactions, we have derived effective spin Hamiltonians that include Heisenberg, Kitaev, and other anisotropic couplings in the strong-coupling limit as a result of a suitable manipulation of interfering paths in the second-order kinetic exchange processes. The model allows for controlling the relative strength

of these interactions through the tuning of the different microscopic tunneling strengths. This type of control is the characteristic feature of cold-atom quantum simulators, suggesting that it might be possible to study the emergence of spin-liquid phases and the influence of different coupling regimes in these quantum platforms in the near future.

We have indeed derived a specific proposal that builds on current progress in Raman optical-lattice platforms, which have recently allowed to engineer synthetic spin-orbit coupling in ultra-cold atomic systems with an increasing precision and control [67–70]. These advances enable the design of complex, direction-dependent tunneling matrices in optical lattices, which are key ingredients for generating the bond-anisotropic interactions of the Kitaev type, as well as to control the relative strength of additional perturbations that are key in the field of Kitaev materials [16–18]. While implementing the full scheme proposed here remains experimentally demanding, and would require further progress in site-resolved control and Raman-assisted scheme stability, we anticipate that future work may identify simplified variants or alternative implementations that retain the essential physics while easing experimental requirements.

We have identified specific Raman-lattice configurations where both ferromagnetic and antiferromagnetic Kitaev couplings arise at leading order, offering a distinct perspective compared to solid-state systems where such terms typically result from higher-order effects and cannot be externally manipulated. Moreover, we have discussed how the transition from a semi-metallic to a Mott insulating phase can be traced as the Hubbard interactions are increased, highlighting how spin-liquid physics can emerge from itinerant fermions within a Hubbard-like framework. To gain further insight into the nature of these emergent phases, we analyzed the model in a inter-layer ribbon geometry, which enabled the application of efficient MPS techniques. This allowed us to explore precursors of the two-dimensional spin liquid behavior in a setting that is numerically tractable, while still capturing the essential physics associated with bond-dependent interactions and strong correlations.

Recent experimental advances, such as the Floquet-based [60] realization of the Kitaev honeycomb model using programmable Rydberg atom arrays [62] and superconducting qubits [63], offer a complementary approach. These recent work focuses on a direct digital simulation of the Kitaev spin model, and the preparation and characterization of topological ground states and excitations, including the non-Abelian spin liquid phase. In contrast, the present study emphasizes how such spin models can emerge from a microscopic, fermionic origin, allowing one to investigate the transition from itinerant to localized behavior and the conditions under which spin-liquid phases may arise from electronic degrees of freedom. In the strong-coupling limit, it may allow for benchmarking both analog and digital quantum simulations.

As an outlook, our work has identified a family of Fermi-Hubbard bilayers that yield a playground to explore the emergence of spin liquids. We believe that future efforts may elucidate more detailed properties of this emergence, and not only focus on the properties of the spin-liquid phases, but actually on the universality of the various critical lines that separate the semi-metallic phases form them. Additionally, it would be interesting to see to investigate finite-temperature and finite-density phenomena, real-time dynamics, and generalizations to other lattice geometries.

### ACKNOWLEDGMENTS

A.B. acknowledge support from PID2021- 127726NB-I00 (MCIU/AEI/FEDER, UE), from the Grant IFT Centro de Excelencia Severo Ochoa CEX2020- 001007-S, funded by MCIN/AEI/10.13039/501100011033, from the the CAM/FEDER Project TEC-2024/COM 84 QUITEMAD-CM, and from the CSIC Research Platform on Quantum Technologies PTI- 001. D.G.-C. acknowledges support from the European Union's Horizon Europe program under the Marie Skłodowska Curie Action PROGRAM (Grant No. 101150724).

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
