# Peer review of "Emergent Kitaev materials in synthetic Fermi-Hubbard bilayers"

_SciPost Physics, doi:SciPost Phys. 19, 130 (2025)_

## Round 1 · Referee Report · Anonymous (Referee 1) · 2025-7-4

Strengths

This is a thorough investigation of bilayer Fermi-Hubbard bilayers and ribbons with emergent Kitaev-like models or symmetry protected topological order. The Schrieffer–Wolff transformation is used to derive effective pseudo-spin models, similarly to the celebrated t-J model [K.A.Chao, J.Spałek, A.M.Oleś, Phys. Rev.B 18, 3453 (1978)], with additional interactions on top of the Kitaev coupling that are similar to those encountered in the Kitaev materials. The manuscript also outlines possible ultracold atoms' realization that opens a way towards quantum simulation of the models that are hard to simulate with classical computers either due to the sign problem (Monte Carlo) or too much entanglement (tensor networks) . I support publication of the manuscript in its present form.

Report

This is an exceptionally good submission to SciPost.

Recommendation

Publish (surpasses expectations and criteria for this Journal; among top 10%)

  • validity: top
  • significance: good
  • originality: good
  • clarity: high
  • formatting: excellent
  • grammar: excellent

Author:  Daniel González-Cuadra  on 2025-10-27  [id 5957]

(in reply to Report 1 on 2025-07-04)

We thank both referees for their careful reading of our manuscript and for their positive and constructive comments. We are especially pleased that both referees find our proposal for realizing Kitaev-type physics in cold-atom systems interesting and worthy of publication in SciPost Physics.

---

## Round 1 · Referee Report · Anonymous (Referee 2) · 2025-9-8

Strengths

  1. Suggests a new cold atom setting for implementation the physics of Kitaev model

  2. Various additional terms in the Hamiltonian such as Heisenberg, \Gamma term, DM term emerge naturally from the simple Hubbard model in the strong coupling limit at half-filling.

  3. The authors present the results of their strong-coupling expansion together with the results of DMRG calculations.

  4. The authors present a discussion of a possible experimental cold-atom setup.

Weaknesses

  1. The paper is very lengthy, and it may have been better to use a shorter format .

  2. The authors make claims (such as discussion of "emergence" in the Introduction), and in their discussion of comparison with other works, which seem to understate previous research in this area.

Report

The authors of the paper suggest a quantum simulator, based on a Hubbard model (suggesting their realisation in cold-atom experiments), which provides a realisation (in the strong-coupling limit) of the Kitaev model with additional terms which also are found in the condensed matter candidates for Kitaev physics (e.g. RuCl_3). They show how these additional terms are generated from the Hubbard Hamiltonian, and suggest various regimes of parameters (e.g. for ferromagnetic or antiferromagnetic Kitaev exchanges). In addition they study the phase diagram using DMRG in ladder geometry, and suggest possible experimental implementation.

I find this model quite interesting, and it is also simple, which may allow one to realise and study Kitaev model with additional anisotropies in experiments.

It seems to satisfy acceptance criteria on the point "1. Provides a novel and synergetic link between different research areas." Therefore I recommend the paper for publication with some minor corrections.

Requested changes

  1. Provide an outline and the summary of the main results in the beginning of the paper. Optionally, shorten the paper and move some of the parts to Appendices.
  2. Improve Introduction (see Weaknesses)
  3. Some of the Figures, e.g. 1-4 are very heavy and do not contain much information. Perhaps simplify the figures.
  4. Check spelling

Recommendation

Publish (meets expectations and criteria for this Journal)

  • validity: good
  • significance: good
  • originality: good
  • clarity: ok
  • formatting: excellent
  • grammar: good

Author:  Daniel González-Cuadra  on 2025-10-27  [id 5956]

(in reply to Report 2 on 2025-09-08)

We thank the referee for the careful reading of our manuscript and for considering that it "provides a novel and synergetic link between different research areas". We have addressed the referee's suggestions and revised the manuscript accordingly, and below we answer the referee's points one by one.

  1. Provide an outline and the summary of the main results in the beginning of the paper. Optionally, shorten the paper and move some of the parts to Appendices.

We thank the Referee for this suggestions. We have written an extended summary of results at the end of the introduction to clarify the main messages for potential readers. On the other hand, we have decided to maintain the current structure of the paper and avoid adding additional appendices.

  1. Improve Introduction (see Weaknesses) We thank the Referee for pointing out the seeming understated research in the emergence of Kitaev's quantum spin liquid (QSL) phases from strongly-correlated lattice models of fermions. We agree that this might not have been written with sufficient detail and may lead to a wrong impression.

There are some works, like the seminal papers previously cited as [15,52] which discuss how the quantum compass spin models do in fact arise from a fermionic model in a Mott insulating regime. What we referred to with "emergence" is to set the focus of attention on the possible critical properties when connecting these string-coupling QSL to the non-interacting metallic or semi-metal phases. It is this second question which was, to the best of our knowledge, far less explored (likely because even the strong-coupling physics with all Kitaev-material terms is already complex enough and not entirely understood) and motivated our work.

Following the Referee's suggestion, we have clarified these differences, and also performed a more exhaustive search in the literature for works discussing the emergence of Kitaev's QSL from a semi-metal as one increases interactions. We have indeed found the following papers that use various mean-field techniques to address this emergence for a particular set of parameters in our model, those that connect to a Kitaev-Heisenberg model

S. R. Hassan, P. V. Sriluckshmy, S. K. Goyal, R. Shankar,and D. Senechal, Phys. Rev. Lett. 110, 037201 (2013).

L. Liang, Z. Wang, and Y. Yu, , Phys. Rev. B 90, 075119 (2014).

[25] J. P. L. Faye, D. Senechal, and S. R. Hassan, Phys. Rev. B 89, 115130 (2014).

More recent papers have addressed this model using tensor network techniques, in partiular projected entangled pairs, and auxiliary-field quantum Monte Carlo, identifying qualitative differences with respect to the mean-field predictions

Shaojun Dong, Hao Zhang, Chao Wang, Meng Zhang, Yong-Jian Han and Lixin He, Chinese Phys. Lett. 40 126403 (2023).

F Mohammadi, SM Tabatabaei, M Kargarian, A Vaezi, Phys. Rev. B 110, 214425 (2024)x

We have included these references in the text, and added a paragraph to discuss the corresponding findings.

  1. Some of the Figures, e.g. 1-4 are very heavy and do not contain much information. Perhaps simplify the figures.

We thank the Referee for this suggestion. We would like to note that we already spent a considerable amount of time formatting these figures so that they are rich in content and, yet, visually appealing. We would thus prefer to keep them in the current format.

  1. Check spelling. We checked and corrected spelling mistakes.

---

## Round 2 · Author Response

We thank both referees for their careful reading of our manuscript and for their positive and constructive comments. We are especially pleased that both referees find our proposal for realizing Kitaev-type physics in cold-atom systems interesting and worthy of publication in SciPost Physics. Below we respond in detail to the specific comments raised by Referee 2.
Response to Referee 2
We thank the referee for the careful reading of our manuscript and for considering that it "provides a novel and synergetic link between different research areas". We have addressed the referee's suggestions and revised the manuscript accordingly, and below we answer the referee's points one by one.
- Provide an outline and the summary of the main results in the beginning of the paper. Optionally, shorten the paper and move some of the parts to Appendices.
We thank the Referee for this suggestions. We have written an extended summary of results at the end of the introduction to clarify the main messages for potential readers. On the other hand, we have decided to maintain the current structure of the paper and avoid adding additional appendices.
- Improve Introduction (see Weaknesses)
We thank the Referee for pointing out the seeming understated research in the emergence of Kitaev's quantum spin liquid (QSL) phases from strongly-correlated lattice models of fermions. We agree that this might not have been written with sufficient detail and may lead to a wrong impression.
There are some works, like the seminal papers previously cited as [15,52] which discuss how the quantum compass spin models do in fact arise from a fermionic model in a Mott insulating regime. What we referred to with "emergence" is to set the focus of attention on the possible critical properties when connecting these string-coupling QSL to the non-interacting metallic or semi-metal phases. It is this second question which was, to the best of our knowledge, far less explored (likely because even the strong-coupling physics with all Kitaev-material terms is already complex enough and not entirely understood) and motivated our work.
Following the Referee's suggestion, we have clarified these differences, and also performed a more exhaustive search in the literature for works discussing the emergence of Kitaev's QSL from a semi-metal as one increases interactions. We have indeed found the following papers that use various mean-field techniques to address this emergence for a particular set of parameters in our model, those that connect to a Kitaev-Heisenberg model
S. R. Hassan, P. V. Sriluckshmy, S. K. Goyal, R. Shankar,and D. Senechal, Phys. Rev. Lett. 110, 037201 (2013).
L. Liang, Z. Wang, and Y. Yu, , Phys. Rev. B 90, 075119 (2014).
[25] J. P. L. Faye, D. Senechal, and S. R. Hassan, Phys. Rev. B 89, 115130 (2014).
More recent papers have addressed this model using tensor network techniques, in partiular projected entangled pairs, and auxiliary-field quantum Monte Carlo, identifying qualitative differences with respect to the mean-field predictions
Shaojun Dong, Hao Zhang, Chao Wang, Meng Zhang, Yong-Jian Han and Lixin He, Chinese Phys. Lett. 40 126403 (2023).
F Mohammadi, SM Tabatabaei, M Kargarian, A Vaezi, Phys. Rev. B 110, 214425 (2024)x
We have included these references in the text, and added a paragraph to discuss the corresponding findings.
- Some of the Figures, e.g. 1-4 are very heavy and do not contain much information. Perhaps simplify the figures.
We thank the Referee for this suggestion. We would like to note that we already spent a considerable amount of time formatting these figures so that they are rich in content and, yet, visually appealing. We would thus prefer to keep them in the current format.
- Check spelling.
We checked and corrected spelling mistakes.

---

## Round 2 · List of Changes

- Added extended summary of results at the end of the introduction.
- Added more references in the introduction.
- Checked spelling.

---

## Editorial Decision

published